# SMT-Learner: Movement Trajectory Learning to Decode Motor Control Strategies

## Abstract

Spatiotemporal movement trajectory (SMT) representation is essential to understanding the motor skill learning and adaptation strategies that inform neurorehabilitation practices. Movement performance metrics (i.e., speed, accuracy) are insufficient to characterize motor control strategies and learning patterns, particularly in individuals with disordered movement. Motor skill learning patterns require an interpretable sequential SMT representation that preserves spatial, temporal, and performance variables. We present a novel SMT-Learner with transformer autoencoders that optimize performance-aware contrastive and adaptive transfer losses, combining cross-task and cross-subject transfer paradigms. SMT-Learner encodes trajectories into a high-dimensional latent space and enables motor performance-aware learning. We introduce an Exploration-Exploitation (E-E) analytical framework that quantifies motor skill learning and control strategies to balance different movement patterns and micro-adaptation. We tested and validated the SMT-Learner with two visuomotor reaching datasets: (1) a prospectively obtained cohort of term and preterm children's motor learning and performance of unimanual and bimanual tasks, and (2) extensively overtrained non-human primates performing target-directed reaching movements. Our ablation and baseline comparison across geometric, statistical, and clustering metrics demonstrated that SMT-Learner outperformed with the lowest reconstruction error (0.086) and optimized clinical correlation with motor performance variables. Investigated E-E patterns significantly correlated with the early and late stages of motor learning and speed-accuracy trade-offs principles. The SMT-Learner framework provides an efficient computational approach to quantify motor learning strategies; potential advanced downstream applications in developmental assessment, neurorehabilitation monitoring, and movement optimization in robotics or brain-computer interfacing.

## 1 Introduction

Recent research in developing analytic tools for motion and kinematic data has applied ML/AI methods to understand motor recovery patterns and prognosis in individuals undergoing neurorehabilitation (i.e., children with cerebral palsy Rapuc et al. (2024), traumatic brain injury Uparela-Reyes et al. (2024); Balaji et al. (2023), post stroke survivors Campagnini et al. (2022)) Choo & Chang (2022); Butepage et al. (2017); Song et al. (2017); Reinkensmeyer et al. (2016). Spatiotemporal movement analysis has created new opportunities to study human motor behavior Wulff et al. (2019); Renso et al. (2013), specifically in movement patterns Viviani & Terzuolo (1982); Kalayeh et al. (2015); Wulff et al. (2019); Long & Nelson (2013), motor rehabilitation Kitago & Krakauer (2013); Levin et al. (2010); van Andel et al. (2008); Murphy et al. (2011), and its underlying neural correlates Svoboda & Li (2018); Gallego et al. (2018). For example, ML-based kinematic analysis using spatial van Andel et al. (2008) or temporal Murphy et al. (2011) parameters of upper extremity tasks can predict movement smoothness and track movement quality. How different motor control strategies are related to upper extremity performance over long-term practice of a motor rehabilitation task is still challenging to decode. Understanding the motor behavior and learning processes from high-repetition and high-density spatiotemporal movement data necessitates new representation learning to decode the patterns. Research in motor learning and development involving spatiotemporal movement trajectories (SMT) utilizes diverse data capture and measurement technologies, including marker and markerless 3D motion capture systems Menolotto et al. (2020), wearable inertial

measurement units (IMUs) Zhou & Hu (2008), that provide high spatial-temporal resolution for precise movement tracking. Digital tablets (e.g., iPads) have recently emerged as powerful tools to capture spatiotemporal aspects of movement, particularly in handwriting, individual finger movement, bimanual coordination, and fine motor skills Palmis et al. (2019); Mia et al. (2024). Importantly, these devices can capture high-resolution temporal and spatial data, including position coordinates, time, velocity, and acceleration during movement execution, needed for motor learning analyses. There are several ML/DL models extensively applied to spatiotemporal trajectory and motion analysis, such as motor recovery prediction Campagnini et al. (2022); Vu et al. (2018) or gait recovery Prakash et al. (2018); Hor et al. (2023), robotics Finn et al. (2016); Saveriano et al. (2023), pedestrian movement analysis Alahi et al. (2016); Rudenko et al. (2020), and autonomous vehicles Schwarting et al. (2018); Maqueda et al. (2018); Kuutti et al. (2020). However, SMT analysis in motor learning studies requires different approaches to decode motor control strategies, micro-adaptation, and learning progress, which potentially impact clinical intervention.

In our recent investigation on motor skill learning and performance using an iPad-based gami-fied visuomotor task among term and preterm school-aged children (N=72, Ages 5-8 years), a new computational problem was identified while interpreting control strategies, due to the nature of non-linear movement dynamics. Compared to term-born children, preterm children have a significantly higher risk of motor delays, which affects their ability to learn and perform motor skills compared to term-born peers Foulder-Hughes & Cooke (2003); Uusitalo et al. (2020); Patel (2016); Allotey et al. (2018); Carter & Msall (2018); Spittle et al. (2016). In addition to lower motor performance, preterm children's ability to learn new motor skills may be impacted due to maladaptive developmental patterns Ortinau & Neil (2015) and differences in brain structures important for sensorimotor function Liu et al. (2010); Adams et al. (2010); Shimony et al. (2016). However, the underlying motor learning strategies used by preterm and term children are difficult to interpret from conventional motor performance parameters. Indeed, there is a distinction in movement variation and adaptation between these two groups. Figure 1 exemplifies an individual's motor learning, where low (during practice) and high (retention) cumulative success rates, the probability of reaching the target at least once over a series of independent trials, had nearly similar movement path lengths. Therefore, a research gap exists in understanding motor learning progress and control strategies from movement data and performance variables.

Existing DL-based trajectory autoencoder and embedding methods, such as STTraj2Vec Zhu et al. (2024), Variational Auto-Encoders (VAEs) Ivanovic et al. (2020), Sequence-to-Sequence Auto-Encoders Sarkar & Ghose (2018); Wang et al. (2022), while effective in movement prediction and classification yet

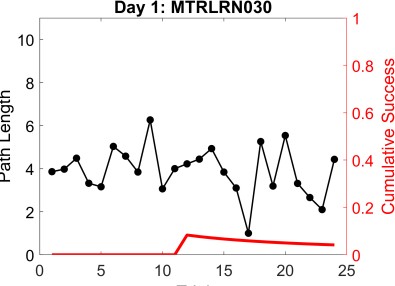 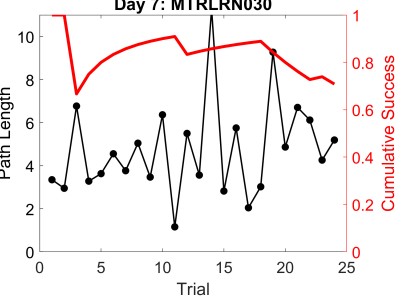

(a) Participant 030 had a very low success rate (red curve) relative to a moderate level of accuracy (4 times ideal path length)

(b) After training, success rate increased to greater than 70% but the accuracy remained at a similar (and perhaps greater) level

Figure 1: Example of a participant's motor skills learning from Day 1 to Day 7, while a traditional parameter (i.e., movement path length) could not capture learning or overall performance on a task. In this task, participants moved a joystick up and down to map movement on a 2D game scene to achieve a target-directed destination from a source.

challenging to interpret complex non-linear relationships in movement patterns. Transformer architectures with self-attention mechanisms Shaw et al. (2018) and self-supervised pre-training approaches (i.e., TimeContrast Guo et al. (2022), MovementContrast Shah et al. (2023)) are capable of capturing sequence dependencies and temporal relationships. However, the repetitive task-based motor training and therapy in rehabilitation practices require more sophisticated methods, which will

preserve trial-to-trial performance variables along with temporal and spatial patterns. This problem motivated the design of a new SMT representation learning framework.

We propose a novel SMT-Learner that combines joint learning with movement performance-aware multi-contrastive loss and adaptive transfer learning. A new human SMT dataset ($D_1$) was created from prior motor skill learning and performance investigation to train and evaluate the model. To cross-validate the generalized applicability of SMT-Learner, evaluate with another hand reaching trajectory dataset ($D_2$) of highly trained non-human primates Scott et al. (2001); Scott & Kalaska (1997). Moreover, we introduce an exploration-exploitation (E-E) analytical framework to quantify motor control strategies and micro-adaptation from the representation, categorized as i) exploratory strategy Svoboda & Li (2018) – where current movement does not correlate with previous movement attempts, and ii) exploitative strategy Gallego et al. (2018) – where prior movements predict current movement. We assessed how movement exploration and exploitation differed between: a) two types of hand movements (unimanual and bimanual), b) term and preterm children, and c) early and late motor learning phases. To provide further SMT-Learner interpretability, we conducted a case study analysis showing two distinct optimal strategies captured by the framework: (1) Curvature optimization to near-straight paths, and (2) Stepwise rectilinear movements with right-angle directional changes.

**Neuroscientific Foundation of E-E Framework:** Exploitation/exploration are well-established concepts to study human and other species' cognitive and motor learning evaluation. E-E frameworks Wyatt et al. (2024) found useful for studying how humans make decisions with known outcomes versus acquiring new information and new outcomes with less certainty. For example, children tend to use more explorative strategies early in development to gather more information, even when this approach may be less rewarded Blanco & Sloutsky (2024). Human visual exploration studies demonstrated Bayesian optimal foraging models Cain et al. (2012) and uncertainty reduction mechanisms Mirza et al. (2018) that are parallel to movement exploration/exploitation. Established principles of motor learning through adaptive combination of motor primitives Thoroughman & Shadmehr (2000) and complementary roles of neural circuits Doya (2000) support E-E mechanisms in biological motor systems. In non-human motor learning studies, E-E concepts are significantly applied to understand motor learning behaviours and neural dynamics. One rodent exploratory behavior study Mumby et al. (2002) demonstrated corticostriatal dynamics that reinforce the reduction of movement variability in repetitive motor skill learning Dhawale et al. (2017) and refinement of muscle synergies Santos et al. (2015), supporting distinction in early exploration and late exploitation strategies. Moreover, this principle also explained how young songbirds produce highly variable vocalizations and strategically transition to stereotyped songs with vocal motor learning Ölveczky et al. (2005); Kojima et al. (2018).

We statistically validated the following hypotheses to demonstrate our framework's effectiveness and its clinical implications. **Hypothesis 1a:** *Early learning will be more explorative and will shift to an exploitative strategy in the late learning phase in all participants.* **Hypothesis 1b:** *Preterm children will exhibit a higher exploration/exploitation (E-E) ratio than term children, particularly for the bimanual skill learning task.* Our cross-validation hypotheses are: **Hypothesis 2a:** *As monkeys were extensively overtrained ($D_2$), their overall E-E ratio will be significantly lower than that of a human learner on an untrained task.* **Hypothesis 2b:** *The E-E ratio will decrease over sequential trials of the same motor learning task, even in well-trained non-human primates, reflecting a micro-adaptation learning process.* The methodological validation will also confirm a speed-accuracy trade-off principle of motor skill development Plamondon & Alimi (1997); Spieser et al. (2017); Molina et al. (2019) preserved by SMT-Learner representation. **Hypothesis 3a:** *Movement performance variables such as movement speed or accuracy will correlate negatively or positively, respectively, with E-E ratio.* This hypothesis will clinically validate our framework's relationship to conventional motor performance variables. Finally, we discuss the potential of the presented approach for clinical translation with limitations and future directions.

## 2 PRELIMINARY

**Movement path.** A real movement path $P$ is a continuous function of time mapping 2D spatial coordinates. Movement path is a function defined as, $P : [0, T] \rightarrow \mathbb{R}^d$, where $T$ is the total time duration of a movement path and for each time point $t \in [0, T]$, $P(t) = (x_t, y_t)$, return a 2D position with x-coordinate and y-coordinate value in the movement space.

**Trajectory.** A trajectory ($\mathcal{T}$) of a moving object is a sequence of positions over time in the movement space, define as $\mathcal{T} = \{(x_1, y_1, t_1), (x_2, y_2, t_2), \ldots (x_n, y_n, t_n)\}$. Where $(x_i, y_i)$ represents spatial coordinates at time $t_i$ with $0 = t_1 < t_2 < \ldots t_n = T$ and n is the number of recorded positions.

**Problem Formulation.** Given a dataset of N spatiotemporal movement trajectories, D $= \{\mathcal{T}_1, \mathcal{T}_2, \ldots, \mathcal{T}_N\}$, where each trajectory $\mathcal{T}_i$ defined as $\mathcal{T}_i = \{(x_j, y_j, t_j) | j = 1, 2, \ldots m\}$. Each trajectory has associated temporal metadata $M_i = \{m_1, m_2, \ldots, m_k\} \subseteq \{pid, task, c\_time, rmsd, is\_success\}$. Here, $c\_time$ is the total completion time of the movement from source to destination in seconds, $task$ indicates experimental visuomotor/movement task, $rmsd$ is root mean square deviation of the original movement path from direct straight line ($source \rightarrow destination$), and $is\_success$ is a flag (0 or 1) that indicate the successfully reaching the destination. We aim to train a trajectory autoencoder to learn a mapping function $f_\theta : \mathcal{T} \rightarrow \mathbb{R}^d$ that transforms each variable-length trajectory into a d-dimensional vector, $\varepsilon_i = f_\theta(\mathcal{T}_i) \in \mathbb{R}^d$ and captures spatio-temporal patterns with preservation of movement performance metrics. We focus on developing SMT-Learner, combining a self-attention encoder and self-supervised pre-training to optimize trajectory reconstruction and movement performance-aware multi-contrastive loss, enabling transfer learning. The goal is to achieve embedding $\mathbb{R}^d$ as a representation of SMT to conduct downstream experiments, specifically the motor learning behavior and the detection of control strategies using E-E analysis.

## 3 METHODOLOGY

SMT-Learner builds upon transformer-based sequential processing Vaswani et al. (2017) and self-supervised contrastive learning Chen et al. (2020), which includes movement performance meta-criterions as contrastive loss for representing trajectories into the embedded space and enables transfer learning Zhang & Gao (2022). SMT-Learner is driven by motor learning principles, designed to learn domain-agnostic representations of planar reaching tasks to decode motor learning and control strategies—measured through speed, accuracy, and success.

### 3.1 TRAJECTORY PROCESSING

**Normalization.** A normalized trajectory $\mathcal{T}'$ is a standardized representation of spatial curve that resolves the variable-lengths and geometric constraints of randomize start and target of a moving object. A trajectory transformation process $\mathcal{N}$ applied to normalize a trajectory, $\mathcal{T}' = \mathcal{N}(\mathcal{T}) = \{(P_j, t_j) | j = 1, 2, 3 \ldots, m\}$, where $\mathcal{T}'$ origin-centered at $\mathcal{T}'_1 = (P_{(0,0)}, t_1)$, target-aligned at $\mathcal{T}'_m = (P_{(0,0)}, t_m)$. transformation process $\mathcal{N}$ involves:

(i) Translate position P of the trajectory to center: $P'_i = P_i - P_1 = \{x_i - x_1, y_i - y_1\}$,

(ii) Rotation by $\theta$ angles to align with target position: $R(\theta) = \begin{vmatrix} x_i \cos\theta - y_i \sin\theta \\ x_i \sin\theta + y_i \cos\theta \end{vmatrix}$, and

(iii) Trajectory is scaled by factor $s$ to finalize position into a specific magnitude: $s = \frac{\left\| \overrightarrow{V}_{target} \right\|}{\left\| \overrightarrow{V}_{end} \right\|}$, where $\overrightarrow{V}_{target} = P_{(0,1)} - P_{(0,0)}$ and $\overrightarrow{V}_{end} = P_n - P_1$. Finally, positional normalization of trajectory is transformed by $\mathcal{T}_{norm} = P'_i \times R(\theta) \times s$.

The rotating/scaling trajectories to a canonical frame removes absolute direction and can obscure biomechanical/cognitive asymmetries. We kept this normalization to simplify trajectory learning while preserving spatial and temporal structure, but we have incorporated directional semantics by adding the target direction angle, $\theta_i = atan2(P_n - P_i)$ to each timestep input and optionally the rotation angle used in normalization as auxiliary inputs.

**Resampling.** For each normalized trajectory sequence with a given length *n*, we applied a parameterized approach to get fixed *m* points that preserve spatial and temporal characteristics. A uniform space parameter, $u'_j = \frac{j-1}{m-1}, for j = 1, 2, 3, \ldots, m$ is defined to obtain exactly $m$ resampled points by identifying segments in the original trajectory where $u_i \leq u'_j < u_{i+1}$, where $u_i = \frac{i-1}{n-1}$, for

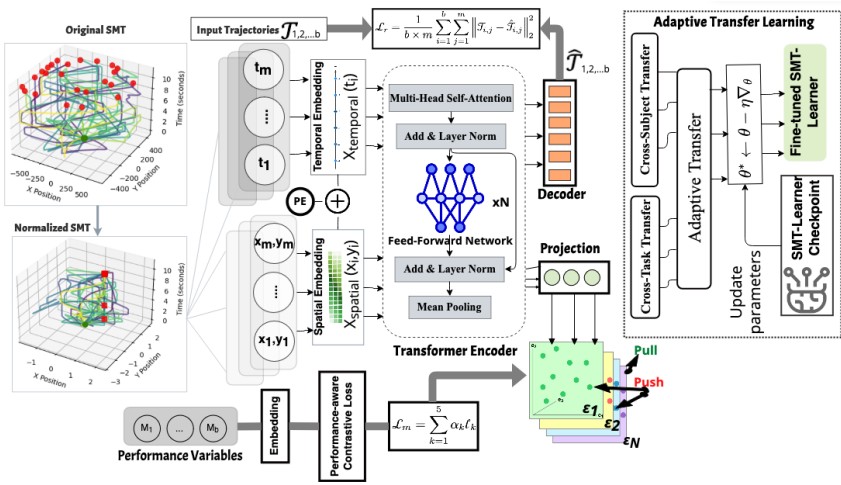

Figure 2: Architecture of SMT-Learner with transfer paradigm

$i = 1, 2, 3, \ldots, n$. An interpolation weight, $\alpha_j = \frac{u'_j - u_i}{u_i + 1 - u_i}$ is used to calculate each dimension of the trajectory using the following equations: $x'_j = (1 - \alpha_j) x_i + \alpha_j x_{i+1}$, $y'_j = (1 - \alpha_j) y_i + \alpha_j y$, and $t'_j = (1 - \alpha_j) t_i + \alpha_j t_{i+1}$.

### 3.2 SMT-LEARNER AND ADAPTIVE TRANSFER

The SMT-Learner consists of five layers (Figure 2): i) dual-stream spatial and temporal embedding, ii) transformer encoder with multi-head self-attention and feed-forward network, iii) dual-headed output with projection and decoder, iv) a movement performance-aware contrastive learning with transfer paradigm, and v) a joint optimization with contrastive and reconstruction loss. The spatial $(x_i, y_i)$ and temporal $(t_i)$ components of each point in the normalized trajectory $\mathcal{T}$ are projected into a $D$-dimensional space using $X_{spatial}$ and $X_{temporal}$ linear transformer with a positional encoder. Spatial & temporal embedding results a tensor $X$ of shape $(b \times m \times D)$, where b is the batch size and m is the number of points in a trajectory, which is the input of the Transformer encoder. Two parallel branches processed the output of the transformer encoder to generate the final embedded representation and reconstructed trajectory using a Projection Head and Decoder, respectively. Embedded output of the non-linear Projection Head He et al. (2020) is $E = ReLU \left( w_1^{proj} . Z_{global} + b_1^{proj} \right) w_2^{proj} + b_2^{proj}$, with shape $(b \times d)$ contains $E \subset (\varepsilon_1, \varepsilon_2, ..\varepsilon_b)$ embeddings where each $\varepsilon_i \in \mathbb{R}^d$. Embeddings $\varepsilon_i$ was used for constative loss calculation. The Decoder reconstructed the original trajectory as $\hat{\mathcal{T}} = Reshape \left( w^{dec} . Z_{global} + b^{dec} \right)$. The reconstructed trajectory used to calculate the reconstruction loss $(\mathcal{L}_r)$ using Equation 1.

$$\mathcal{L}_r = \frac{1}{b \times m} \sum_{i=1}^{b} \sum_{j=1}^{m} \left\| \mathcal{T}_{i,j} - \hat{\mathcal{T}}_{i,j} \right\|_2^2 \tag{1}$$

#### 3.2.1 PERFORMANCE-AWARE CONTRASTIVE LEARNING

The model learns the representation in embedded space $\varepsilon \in \mathbb{R}^d$ from the motor performance meta criterion $(M_i)$ using "pull" and "push" operations, where pull similar trajectories or push dissimilar ones based on the multi-contrastive loss function $(\mathcal{L}_m)$ as calculated using Equation 2 and 3.

$$\mathcal{L}_m = \sum_{k=1}^{5} \alpha_k \ell_k \tag{2}$$

$$\ell_k = \frac{\sum_i \sum_{j \neq i} \psi_k(i, j) \log \frac{e^{\varepsilon_i \cdot \varepsilon_j / \tau}}{\sum_{l \neq i} e^{\varepsilon_i \cdot \varepsilon_l / \tau}}}{\sum_i \sum_{j \neq i} \psi_k(i, j) + C} \tag{3}$$

Where $\ell_k$ is loss components for corresponding meta criterion $m_k$ and weight factor $\alpha_k$ as $\sum_{k=1}^{k} \alpha_k = 1$. $\mathtt{C} = e^{-6}$ to avoid numerical instability. For each batch of embedded trajectories, E, contrastive loss components $\ell_k$ are computed based on the similarity matrix ($\psi_k$) for meta criterion k with a temperature parameter $\tau$ Wang & Isola (2020). Trajectory meta criterion completion time (*c_time*), root mean square deviation (*rmsd*) and successfully reaching the destination (*success*) have been used as specialized similarity measures. Let's define the rmsd distance as d, while the similarity between two trajectories $\varepsilon_i$ and $\varepsilon_j$ from batch $E$ is computed by Equation 4.

$$\psi_{rmsd}(i,j) = 1 - \frac{|d_i - d_j|}{\max_{k,l} |d_k - d_l| + \mathtt{C}} \tag{4}$$

Here, $\max_{k,l} |d_k - d_l|$ find the max difference in sequential paired samples of $E$. Completion time contrastive loss captured comparable timeframe patterns to pull or push embeddings based on the similarity calculation. Other two similarity matrices, $\psi_{c\_time}(i,j)$ and $\psi_{success}(i,j)$ capture movement speed and efficiency to learn representation. Finally, the participant ID ($pid$) and movement tasks ($task$) information were used as cross-subject and cross-task knowledge transfer to balance learning with specific and generalized patterns.

### 3.2.2 Adaptive Learning with Cross-Task and Cross-Subject Transfer

The characteristics of movement trajectory in rehabilitation or robotics space depend on the task executed, which impacts the trajectory shape, such as opening a door or moving an object from source to destination using only up-down, left-right actions. Cross-task knowledge transfer is important to preserve task-specific information and movement patterns in the representation space Shi et al. (2023). Whereas, the cross-subject transfer paradigm allows flexible control on subject-specific knowledge learned across all other subjects, for a target subject to generalize the learning in offline mode. Our transfer process simultaneously optimized joint losses $\mathcal{L}_{total} = \mathcal{L}_r + \mathcal{L}_m$. For a transfer paradigm (i.e., cross-task, cross-subject), two hyperparameters ($\lambda_1$ and $\lambda_2$) with a transfer-specific regularization are applied to optimize loss and appropriate separation between different subjects and tasks. Equations 5 and 6 update weights for a specific transfer type, where $\text{sim}(\varepsilon_i, \varepsilon_j) = \frac{\varepsilon_i \cdot \varepsilon_j}{\|\varepsilon_i\| \|\varepsilon_j\|}$ represents cosine similarity between embeddings and $\mathbb{I}[\text{factor}_i \neq \text{factor}_j]$ is an indicator function for different tasks or subjects, respectively.

$$\mathcal{L}_{\text{transfer}} = \mathcal{L}_r + \lambda_1 \mathcal{L}_m + \lambda_2 \mathcal{L}_{\text{regularization}} \tag{5}$$

$$\mathcal{L}_{\text{regularization}} = \frac{1}{N} \sum_{i=1}^{N} \sum_{j \neq i} \max(0, \text{margin} - \text{sim}(\varepsilon_i, \varepsilon_j)) \cdot \mathbb{I}[\text{factor}_i \neq \text{factor}_j] \tag{6}$$

However, motor learning is intrinsically individualized and context-dependent Shmuelof et al. (2012). Inter-subject variability and task-specific complexity require different control strategies. Static weight transfer may reduce individual differences Long et al. (2015); Kendall et al. (2018), necessitating dynamic weight updates to capture motor signatures and knowledge transfer between participants and tasks. We combined both paradigms with an adaptive transfer mechanism Cao et al. (2010), which updates model parameters $\theta^* \leftarrow \theta - \eta \nabla_\theta$ using Equation 7.

$$\mathcal{L}_{\text{adaptive}}^{(t)}(\theta) = \mathcal{L}_r + \lambda_1 \cdot \sum_{k=1}^{5} \hat{\alpha}_k^{(t)} \cdot \mathcal{L}_k(\theta) + \lambda_2 \cdot \mathcal{L}_{\text{regularization}} \tag{7}$$

Performance-aware multi-contrastive loss components, $\sum_{k=1}^{5} \hat{\alpha}_k^{(t)} \cdot \mathcal{L}_k(\theta)$ represent the core adaptive weighting mechanism dynamically balanced transfer context. During training, time-dependent weights $\hat{\alpha}_k^{(t)}$ adjust based on improvement rates from loss history windows. Transfer-specific modulation factors emphasize different components based on whether knowledge is transferred across subjects or tasks.

### 3.3 Exploration-Exploitation Analytical Framework

We introduced a quantitative method, the Exploration-Exploitation (E-E) framework, to analyze the decoded learning patterns and control strategies from the SMT-Learner representation. In the motor

Table 1: Summary of SMT-Learner pretraining/finetuning results, all experiments conducted on $D_1$

| Paradigm | Pretrain | Evaluate (target) | Zero-shot mean [95% CI] | Fine-tuned mean [95% CI] | $\Delta\%$ |
|---|---|---|---|---|---|
| Exp1 | D1 | D1 test | 1.55 [1.525, 1.575] | 1.00 [0.98, 1.02] | $-35.5\%$ |
| Exp2 | D1 Unimanual | D1 Bimanual | 1.10 [1.08, 1.12] | 0.55 [0.541, 0.559] | $-50.0\%$ |
| Exp3 | D1 Term | D1 Preterm | 1.05 [1.041, 1.059] | 0.45 [0.441, 0.459] | $-57.1\%$ |
| Exp4 | D1 Unimanual + Term | D1 Bimanual + Preterm | 1.05 [1.041, 1.059] | 0.12 [0.111, 0.129] | $-88.6\%$ |

skill learning process, participants learn mastery of a task by repetition. Exploration scores measure movement diversity, and exploitation scores measure how prior movement predicts current movement. Exploration$(\varepsilon_i) = \min_{j<i} \text{Dist}(\varepsilon_i, \varepsilon_j) \times (\beta_1 + \beta_2 e^{-i\alpha})$, where, $\alpha$ is decay factor for trial sequence and $\beta_1, \beta_2$ are weights for movement novelty and trial sequence. The exploitation score measures how prior movement is predicting current movement using a window size $(W_i)$ and a similarity matrix, Exploitation$(\varepsilon_i) = \frac{1}{|W_i|} \sum_{j \in W_i} \text{Sim}(\varepsilon_i, \varepsilon_j)$. Finally, E-E Ratio $= \frac{\text{Exploration}(\varepsilon_i)}{\text{Exploitation}(\varepsilon_i)}$, consider as a factor of sequential motor learning. We applied MIN distance (minimum Euclidean distance in embedding to any prior trial within a decayed window) and KNN algorithm with $W = 120$, $\alpha = 0.05$, $\beta_1 = 0.10$, and $\beta_2 = 0.90$, validated via average distance and density-based novelty. Three consistent patterns supported the selection of the optimized hyperparameters to compute E–E metrics. Sensitivity and clustering analyses are detailed in Appendix A.3.

## 4 RESULTS & DISCUSSION

SMT-Learner optimized all loss components in the pretraining stage to learn generalizability from the domain data (Appendix Section A.1 DATASETS). In the transfer stage, the SMT-Learner pre-trained model was fine-tuned using $D_1$ to update parameters based on the transfer paradigms (cross-task, cross-subject, and adaptive transfer). The complete experimental setup and transfer experiments are detailed in Appendix Section A.2 EXPERIMENTAL SETUP. We computed 5 seeds with mean ±95% confidence intervals (t-based, df=4) for all SMT-Learner transfer experiments (Exp1-Exp4), reported transfer loss

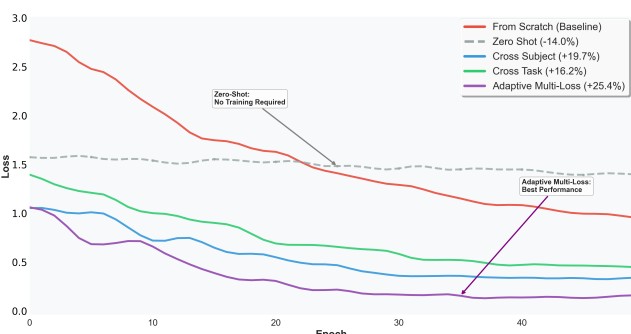

Figure 3: Comparison of adaptive transfer learning with the SMT-Learner baseline and other paradigms

($\mathcal{L}_{\text{transfer}}$) in Table 1. Adaptive transfer loss ($\mathcal{L}_{\text{adaptive}}^{(t)}(\theta)$ with multi-temporal components dropped significantly (overall 25.4% performance improvement) compare to the SMT-Learner baseline model (Figure 3). Held-out evaluations were performed on $D_2$ tasks/sessions never seen during training to confirm cross-dataset generalization. $D_1 \rightarrow D_2$ zero-shot overall loss dropped 1.55 to 1.24 and 1.28 on a single task held-out samples ($D_2$ Experimental Task 1). Using the $D_1$ Preterm finetuned checkpoint (no $D_2$ pretraining/finetuning), the loss dropped to $\sim$1.18. Finally, adaptive transfer fine-tune loss reaches 0.98, evidence that SMT-Learner captures transferable motor structure rather than dataset-specific regularities and provides a scale-stable E–E metric (Appendix A.5).

### 4.1 STATISTICAL TESTING & HYPOTHESIS VALIDATION

We tested Hypotheses 1a/1b on $D_1$ using a three-way ANOVA (cohort, phase, task) with E–E ratio as the dependent variable. Early→late learning showed a robust shift from exploration to exploitation (F=343.1, p<0.001, $\eta^2$=0.050), with E–E decreasing from 0.667±0.056 to 0.163±0.020 (Fig. 4a, 4b). Preterm children exhibited higher E–E than term (0.444±0.059 vs. 0.386±0.030; Cohort: F=3.72, p=0.054), partially supporting 1b. Task effects were significant: bimanual > unimanual in both term (0.286±0.022 vs. 0.255±0.019; p=0.033) and overall (0.469±0.049 vs. 0.361±0.040; F=10.43,

p=0.001), consistent with delayed bimanual coordination in preterm Cooke & Foulder-Hughes (2003); Schneider et al. (2008).

For Hypotheses 2a/2b, we compared human $D_1$ E-E ratio to overtrained monkey $D_2$ reach trajectories: human E–E 0.2823±0.0128 vs. monkey 0.0542±0.0046 (t=39,957, p<$10^{-4}$, $d$=0.354), strongly supporting 2a (Fig. 4c). Block-wise slopes (120 trials; bootstrap $N$=5000) yielded near-linear refinement with $R^2 = 1.00$ in both species and distinct learning rates: human $-0.17$ [$-0.19$, $-0.15$] vs. monkey $-0.008$ [$-0.017$, $-0.001$], confirming 2b (Fig. 4d). These results align with literature on motor variability and expertise Spieser et al. (2017) and are summarized in Table 2.

Speed-accuracy trade-offs, widely accepted mathematical concepts in target-directed human movement, are described as neuromuscular synergy during motor execution Plamondon & Alimi (1997); Smyrnis et al. (2000); Spieser et al. (2017). Speed-accuracy trade-offs and statistical correlation between E-E ratio and performance variables (movement speed and accuracy) are illustrated in Figure 4e, 4f, and 4g. We found negative correlations between speed and accuracy in both term (r=-0.40, p<0.0001) and preterm (r=-0.36, p<0.0001) groups. We also found the E-E ratio positively correlated (Term, r=0.5 and Preterm, r=0.45) with the movement speed and negatively correlated (Term, r=-

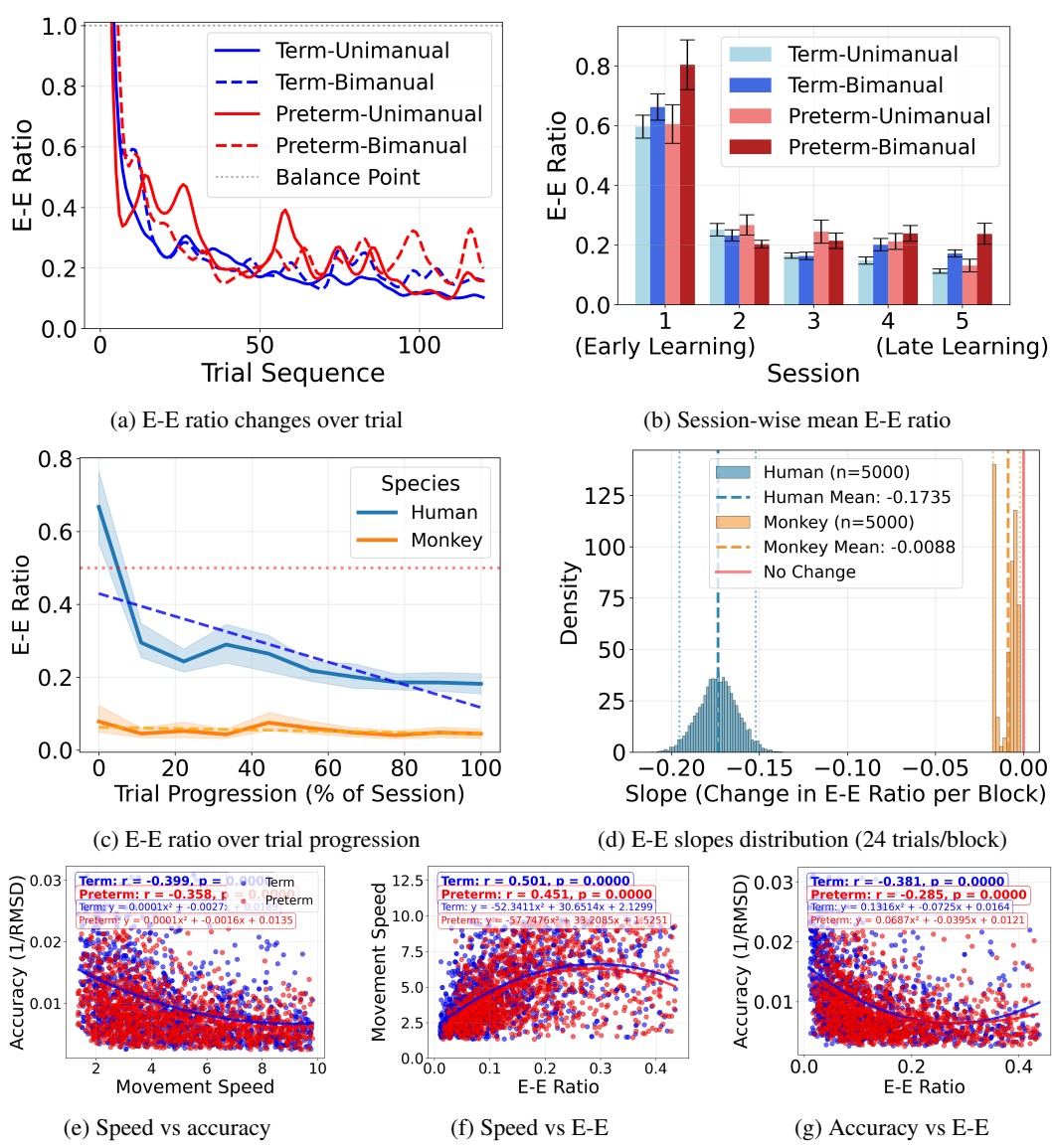

(a) E-E ratio changes over trial

(b) Session-wise mean E-E ratio

(c) E-E ratio over trial progression

(d) E-E slopes distribution (24 trials/block)

(e) Speed vs accuracy

(f) Speed vs E-E

(g) Accuracy vs E-E

Figure 4: Statistical analysis of E-E ratio in different cases: (a-b) E-E ratio changes over trial and session progression, (c-d) E-E dynamics between human learners and monkey, and (e-g) Speed-accuracy trade-offs and correlation between speed/accuracy and E-E ratio

Table 2: Evidence-based cross-matches of SMT-Learner's E–E findings with prior literature

| Hyp. | Prior findings | E–E result (mean $\pm$ 95% CI) | Effect size | Test (t/p) | Interpretation |
|---|---|---|---|---|---|
| 1a | Early→late stabilization in motor learning 
 Smyrnis et al. (2000) | $0.667 \pm 0.056 \rightarrow 0.163 \pm 0.020$ | $d = 0.35$ | $t = 39{,}957;$ 
 $p < 10^{-4}$ | E–E declines with practice; stabilization phase reached. |
| 1b | Term vs preterm adaptability differs 
 Hadders-Algra (2010); Dusing & Harbourne (2010) 
 Ferrari et al. (2012) | Term: $0.386 \pm 0.030;$ 
 Preterm: $0.444 \pm 0.059$ | $d = 0.28$ | $p < 0.01$ | Typical children show lower E–E (more exploitation). |
| 2a | Practice reduces variability (human vs non-human) 
 Mandelblat-Cerf et al. (2009); Dhawale et al. (2017) | Human: $0.2823 \pm 0.0128;$ 
 Monkey: $0.0542 \pm 0.0046$ | $d = 0.35$ | $p < 10^{-4}$ | Both species exhibit reduced variability with training. |
| 2b | Skill refinement continues post stabilization 
 Smits-Engelsman et al. (2020); Churchland et al. (2006) | Slope: $-0.17$ ($R^2 = 1.0$); 
 Monkey: $-0.008$ ($R^2 = 1.0$) | — | $p < 0.05$ | Slow shift toward exploitation; continued refinement. |

Table 3: Performance of SMT-Learner: Ablation study with contrastive and transfer configurations

| Configuration | Performance Metrics | | | | | | | | |
|---|---|---|---|---|---|---|---|---|---|
| | rMSE ↓ | Ep-Err ↓ | Curve-Err ↓ | T-Corr ↑ | R-Corr ↑ | S-Corr ↑ | Traj-C ↑ | Task-C ↑ | Sub-C ↑ |
| **Adaptive Transfer** | **0.086** | **0.072** | **1.577** | **0.893** | **0.539** | **0.970** | **0.720** | **0.550** | **0.038** |
| No Transfer | 0.145 | 0.089 | 1.634 | 0.756 | 0.423 | 0.912 | 0.685 | 0.487 | 0.025 |
| Cross-Task Only | 0.098 | 0.078 | 1.592 | 0.834 | 0.501 | 0.945 | 0.702 | 0.523 | 0.031 |
| Cross-Subject Only | 0.102 | 0.081 | 1.588 | 0.817 | 0.487 | 0.938 | 0.695 | 0.541 | 0.034 |
| No Contrastive | 0.197 | 0.022 | 1.891 | 0.123 | 0.001 | -0.020 | 0.412 | 0.298 | 0.018 |
| $\psi_{c\_time}$ only | 0.086 | 0.093 | 1.568 | 0.479 | 0.289 | 0.191 | 0.713 | 0.548 | 0.037 |
| $\psi_{rmsd}$ only | 0.098 | 0.137 | 1.646 | -0.005 | 0.002 | 0.111 | 0.720 | 0.550 | 0.037 |
| $\psi_{success}$ only | 0.087 | 0.127 | 1.691 | 0.191 | 0.111 | **0.993** | 0.720 | 0.548 | 0.038 |
| $+\theta$ (target direction) | 0.151 | 0.019 | 1.787 | **0.980** | 0.539 | 0.910 | 0.720 | 0.548 | 0.038 |
| $+\theta$+rotation angle | 0.111 | 0.015 | 1.903 | 0.929 | **0.652** | 0.940 | 0.720 | 0.548 | 0.038 |

0.38 and Preterm, r=-0.29) with the movement accuracy. These findings validate our framework's relationship to the clinical assessment of motor performance and captured speed-accuracy trade-offs.

A case study is presented in Appendix Section A.4, where we demonstrate the E-E matric capable of detecting two optimal control strategies (Curvature and Stepwise).

## 4.2 SMT-LEARNER PERFORMANCE EVALUATION

We applied geometric, statistical, and clustering neighborhood analysis to evaluate the quality and characteristics of SMT-Learner representation. Assessment metrics are as follows: (i) Trajectory reconstruction quality: Reconstruction Mean Squared Error (rMSE), Mean Endpoint Error (Ep-Err), and Mean Curvature Error (Curve-Err); (ii) *Statistical correlation with movement performance variables:* Completion time (considered as movement speed) correlation (T-Corr), correlation with the root mean square deviation of movement (considered as accuracy) from the optimal path (R-Corr), and correlation with successfully reaching the target (S-Corr); and (iii) *Clustering neighborhood consistency:* trajectory shape consistency (Traj-C), cross-task consistency (Task-C), and cross-subject consistency (Sub-C).

### 4.2.1 ABLATION STUDY

Our ablation studies validate the necessity and contribution of performance-aware contrastive learning and transfer paradigms. Ablation results in Table 3 show that removing contrastive learning causes an 86% drop in temporal correlation performance, dropping T-Corr from 0.893 to 0.123, and R-Corr from 0.539 to near-zero (0.001). Adaptive transfer significantly improves performance correlations and clustering consistency compared to other transfer paradigms or no transfer. Moreover, adding $\theta$ improves timing and path-accuracy correlations and substantially reduces endpoint error ($0.072 \rightarrow 0.019$ with $\theta$ and 0.015 with $\theta$+rotation angle), but rMSE goes down. Such as, $\theta$ components increase T-Corr +0.087 and R-Corr +0.121; S-Corr remains strong ($> 0.90$). These results indicate that adding target direction as an auxiliary input, along with normalized trajectory, restores asymmetry-related cues and improves performance.

### 4.2.2 BASELINE COMPARISON

Existing trajectory analysis methods lack downstream applicability for motor control and rehabilitation practices Hu et al. (2023); Chen et al. (2024). We selected four methods for comparison

that closely matched study objectives: (1) STTraj2Vec Zhu et al. (2024), (2) Variational Auto-Encoders (VAEs) Ivanovic et al. (2020), (3) Sequence-to-Sequence Auto-Encoders (Seq2Seq) Sarkar & Ghose (2018); Wang et al. (2022), and (4) Trajectory Masked Autoencoders (Taj-MAE) Chen et al. (2023). We found that SMT-Learner outperformed with the best rMSE, Ep-Err, Curve-Err, and S-Corr, in both tests with $D_1$ (training and finetuned) and $D_2$ held-out evaluation (Table 4).

However, STTraj2Vec optimized temporal/spatial continuity without incorporating outcome constraints (success/failure), yielding extremely high T-/R-Corr ($L_2$ norm of embedding grows with time or deviation). In motor tasks, failures or inefficient trials are longer and more deviant. If embedding magnitude amplifies only temporal/spatial characteristics, the same feature that boosts T-/R-Corr inversely relates to success, yielding negative S-Corr. SMT-Learner balanced temporal/spatial fidelity with performance-relevant structure. As a result, it maintains very high positive S-Corr while keeping competitive T-/R-Corr. Baseline comparison with Traj-MAE reflected that the similar studies (i.e., Forecast-mae Cheng et al. (2023), SEPT Lan et al. (2023)) would also fail to perform better in the investigated metrics as they lack performance-aware representation.

Table 4: Baseline comparison results

| Method | rMSE | Ep-Err | Curve-Err | T-Corr | R-Corr | S-Corr |
|---|---|---|---|---|---|---|
| STTraj2Vec | 0.386 | 0.095 | 3.666 | **0.982** | **0.647** | -0.797 |
| VAE | 0.412 | 0.089 | 4.758 | 0.136 | 0.096 | -0.135 |
| Seq2Seq [48,49] | 0.190 | 0.215 | 3.619 | -0.996 | -0.653 | 0.819 |
| Traj-MAE | 0.111 | 0.290 | 28.304 | -0.960 | -0.638 | 0.779 |
| **SMT-Learner**$_{D_2}$ | 0.089 | 0.0944 | 1.867 | 0.735 | 0.522 | 0.9358 |
| **SMT-Learner**$_{D_1}$ | **0.086** | **0.072** | **1.577** | 0.893 | 0.539 | **0.970** |

## 5 LIMITATIONS & FUTURE DIRECTION

SMT-Learner, while effective in capturing spatiotemporal dynamics of trajectory, has several constraints, including datasets, dimensionality, and generalizability. Embedding dimension, similarity thresholds, and sequential window sizes require systematic investigation for different movement trajectories across species, clinical conditions, and learning tasks. Moreover, the behavioral experiments were conducted in 2D space, which can be extended to 3D trajectories with minimal modification. We can simply modify input layer from $\mathcal{T} = \{(x, y, t) \mid$ spatial coordinates + time$\}$ to $\mathcal{T} = \{(x, y, z, t) \mid$ 3D coordinates + time$\}$ and normalizing 3D vector operations for position, rotation, and scaling. E-E analysis depends on embedded spaces and temporal continuity, and may be less sensitive when a learner suddenly shifts strategy, leading to discontinuous skill acquisition Newell (2014). Another limitation is that the findings on the unimanual vs bimanual visuomotor tasks represent a subset of motor skills, as the scope of this study only focused on repetitive motor tasks to understand learning behavior and micro-adaptation. However, other domains, such as gross motor skills, manual dexterity, or force production tasks, may require SMT-Learner fine-tuning using cross-task/cross-subject transfer to analyze E-E dynamics, which will be explored in the future.

## 6 CONCLUSIONS

Existing approaches to analyzing SMT data typically reduce complex motor trajectories to singular spatiotemporal parameters, such as movement accuracy or velocity. While important, this approach loses information about the dynamic nature of the action. Instade SMT-Learner, combined with an exploration-exploitation (E-E) metric to quantify fundamental aspects of motor skill learning across developmental contexts. Our computational & analytical approach bridges AI into neuromotor control, developmental psychology, and neurorehabilitation insights that could inform therapeutic and intervention planning by identifying learning strategy deficits to guide optimal therapy for populations with developmental disorders. Extensive experiments with two real datasets and hypothesis cross-validation revealed fundamental characteristics of skill acquisition, shifting from exploration-dominant to exploitation-dominant strategies over practice. In the future, adaptive transfer learning with data from different motor learning tasks and conditions would improve the capability for personalized therapy and modulate E-E balancing for individual learning profiles.

ETHICS STATEMENT

*Human Subjects Protection.* This study involves human subject data. We collected data of term and preterm-born children ($D_1$) to investigate motor skills learning and control strategies based on the IRB-approved experimental protocol. The parents or guardians of the child (as participants aged 5-8 years old) signed an informed consent form to share non-identifiable data for research purposes. We ensured HIPAA-compliant data storage and removed all identifiable information (e.g., name, date of birth, phone number) from the dataset. We used anonymous identifiers (e.g., MRTLRN###) only.

REPRODUCIBILITY STATEMENT

We supply all requisite materials and documentation to assure the reproducibility of the SMT-Learner framework. The source code implementation of the SMT-Learner architecture, encompassing the adaptive loss weighting mechanism, cross-task and cross-subject transfer learning modules, together with all experimental configurations, is accessible via an anonymous 4open.science repository Anonymous (2025). Furthermore, we have included a supplementary zip file comprising: (1) the complete codebase with README guidelines for environment configuration, data preprocessing, model training, and evaluation methodologies; (2) evaluation scripts that replicate all documented results; and (3) generated results, figures, and graphs.

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

## A   APPENDIX

### A.1   DATASETS

#### A.1.1   $D_1$: HUMAN MOVEMENT DATA.

$D_1$ contains 16320 trajectories of term (73.5%) and preterm (26.5%) born children. Data was collected using an iPad-based visuomotor game, designed for unimanual and bimanual motor learning using controlled psychophysical tasks. We conducted a cross-sectional multi-visitation observation study to assess motor skills learning and performance in term and preterm children aged 5-8 years. This study aimed to measure a child's development and overall abilities to learn new motor tasks and establish causal links between motor learning and performance. We explored the relationship between motor planning and execution networks for completing functional tasks and identified primary contributors to overall motor development. The university's Institutional Review Board (IRB) approved study protocol.

**Study Protocol:** We examined unimanual and bimanual motor learning using controlled psychophysical tasks. We created a straightforward yet challenging visuomotor task that tested how participants learned a new mapping between joystick and cursor movement. The experimental tasks (Figure 5) involve moving a cursor on an iPad 12.9-inch screen (cartoon bee) to a visual target (flower) using a joystick. The mapping of joystick direction to cursor movement systematically var-

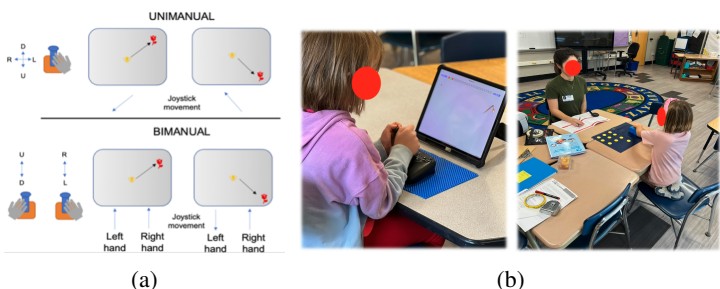

(a)                          (b)

Figure 5: Study protocols and data collection (a) Experimental design of the tasks in an iPad game (unimanual and bimanual tasks, and (b) A session of participants' data collection using the mHealth system in an elementary school networks

ied. For the unimanual task, a single two-dimensional joystick was used with the direction map inverted (e.g., moving the joystick upward moves the cursor downward, and moving the joystick rightward moves the cursor leftward). For the bimanual task, two one-dimensional (vertical movement only) joysticks were controlled with each hand, with the left joystick controlling the cursor vertically and the right joystick controlling the cursor horizontally. The unimanual learning task was a mirror reversal task. Furthermore, the bimanual task involved the non-intuitive 90° rotation of the directionality of one joystick, which was even more challenging. These adaptations, while easy for adults to learn, were challenging for young children. Thus, we propose that the tasks were appropriately complex for the age (5-8 years old) of the participants performing them.

**Task Parameters:** For each trial, the cursor starts in the center of the screen. Six targets within each of the four quadrants of the 2D screen were selected randomly; thus, the participant moved to 24 targets during each practice block. The variability in the initial location of the target should enhance motor learning based on the effects of a variable practice schedule.

Table 5: Participant characteristics

| Characteristics | Term | Preterm |
|---|---|---|
| # of Participant (N) | 50 | 18 |
| Age Group (N, %)
5-6

7-8 | 19 (38.0%)
31 (62.0%) | 7 (38.8%)
11 (61.1%) |
| Gestational Age (weeks), mean $\pm$ SD | 39 $\pm$ 2 | 31 $\pm$ 3 |
| MABC-2 percentile, mean $\pm$ 95%CI | 39.91$\pm$0.75 | 23.61$\pm$0.87 |

The participant has 10 seconds to complete the trial and reach the target. A new trial begins if the cursor does not reach the target in under 10 seconds. Visual feedback on trial success (smiley face) or failure ("Try again" message) was provided. To prevent participants from moving in a unidirectional manner during the bimanual task (i.e., moving only the left joystick to move vertically, then the

right joystick to move horizontally), cursor movement was programmed to advance only when both joysticks are moved. In each trial, we recorded source and target destinations, (x,y) coordinates as continuous movement paths with time dimension at 120 Hz sampling rate.

**Participants & Data Collection:** We collected data from 72 participants, 68 of whom completed all blocks of tasks successfully on Day-1, Day-2, and Day-7. Table 5 shows a summary of the participants' characteristics. Along with the game data, we tested participants' standard battery assessment (MABC-2: Movement Assessment Battery for Children). Among term and preterm children, we found a significant difference in MABC-2 percentiles (23.61±0.87 vs. 39.91±0.75, p<0.001), demonstrating substantial clinical and neurodevelopmental validation. Each of the participants practiced 6 blocks of 24 trials each, completed on Day 1, with 1-2 minutes of rest between each block. To examine retention, a single block of 24 trials was repeated on Days 2 and 7 (retention blocks). A total of 680 blocks/sessions of data was collected with 680x24 trials. This dataset contains 16320 trajectories of term (73.5%) and preterm (26.5%) children, where each task contains 50% the trajectories.

### A.1.2 $D_2$: Non-human Reaching Movement

$D_2$ contains non-human primates' arm reaching trajectories Scott et al. (2001); Scott & Kalaska (1997), a groundbreaking study investigated the neural basis of motor control and hand movement kinematics. Three rhesus monkeys were highly trained to perform horizontal planar reaching movements wearing mechanical exoskeletons. The task was centered on reaching a target arranged in a circle with five experimental conditions (e) and collected spatiotemporal positions, velocity, and joint angles with neural recordings. Each hand trajectory contains (x, y, t) coordinates, matching the expected input format for SMT-Learner. An example of experiment reaching trajectories to uniformly distributed targets at 0, 45, 90, 135, 180, 225, 270, and 315 degrees are illustrated in Figure 6. This dataset includes 16 unique reaching directions with standardized durations (∼576ms). We used a total of 23639 trajectories from a total of 587 sessions, where 75% of the sessions contained 48 trials in four experimental tasks.

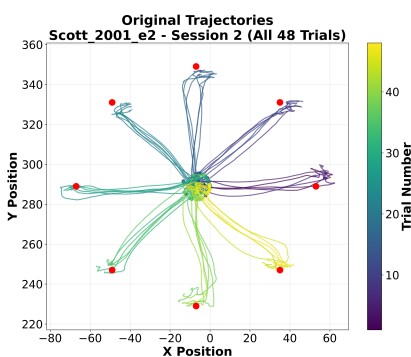

Figure 6: Examples of monkey's hand movement trajectories of e2 experiment

### A.2 Experimental Setup

Experiments were conducted using NVIDIA GH200 Superchips (H100 configured with 80 GB SXM5, 26 vCPUS, 225 GiB RAM and 2.8 TiB SSD). We followed a two-phase training and evaluation approach with two datasets $D_1$ and $D_2$. In our first phase, SMT-Learner was pre-trained using $D_1$ with a 90:10 split ratio for the train and validation partitions, and 32 SMT as the input batch size. The total joint loss combines reconstruction and multi-contrastive objectives as $\mathcal{L}_{total} = \mathcal{L}_r + \mathcal{L}_m$. The model was trained for each component of contrastive loss separately, as well as multi-contrastive loss by combining a weighted function of loss components. With 50 epochs, early stopping was imposed based on validation loss, and the AdamW optimizer was used with a learning rate of 0.0001 Loshchilov & Hutter (2017). We evaluated four experimental conditions in pretraining/finetuning paradigms to separate the cross-task and cross-subject transfer effects on adaptive transfer, as follows.

1. Exp1: Pretrain on $D_1$ → test on $D_1$ → zero-shot on $D_2$ (held-out)

2. Exp2: Cross-Task transfer: Pretrain on $D_1$ Unimanual → Test on $D_1$ Bimanual → Zero-shot on $D_2$ (Experimental task 1: Scott_2001_e1)

3. Exp3: Cross-Subject transfer: Pretrain on $D_1$ (Cohort==Term) → test on $D_1$ (Cohort == Preterm) → Zero-shot on $D_2$

4. Exp4: Adaptive transfer: Combine Exp2 and Exp3 → test on $D_1$ → zero-shot on $D_2$

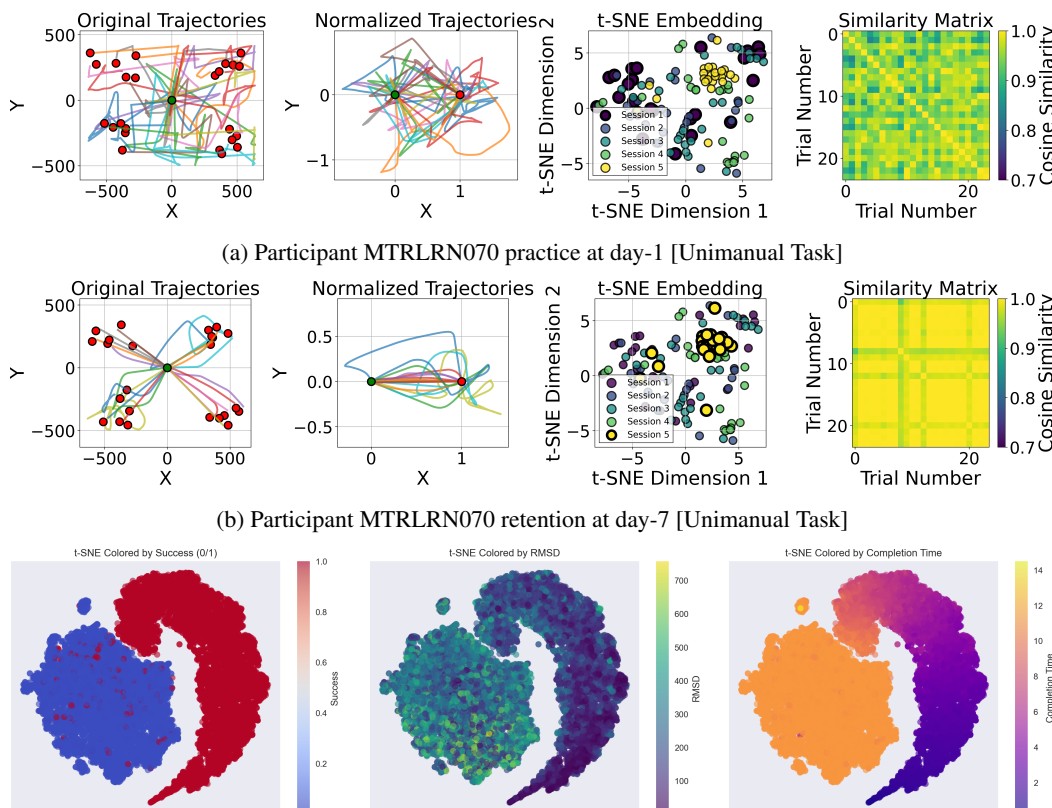

(a) Participant MTRLRN070 practice at day-1 [Unimanual Task]

(b) Participant MTRLRN070 retention at day-7 [Unimanual Task]

(c) 2D tSNE visualization of the embedded space colored by (left to right) Success, RMSD, and Completion time

Figure 7: Trajectory representations in embedding space. (a–b) Embedded layout and trial-by-trial trajectory similarity for a Unimanual participant. (c) t-SNE shows tighter clusters for higher-skill learners (moon shape); unsuccessful trials form a compact "ball" cluster and are associated with longer durations and greater path deviation.

### A.3 SENSITIVITY AND CLUSTERING ANALYSIS

Figures 7a and 7b compare a participant's embedded trial-by-trial trajectory similarity on the first practice day versus Day-7 retention. At Day 7, embeddings exhibit closer, more stable neighborhoods and reduced dispersion, indicating learning adaptivity and a shift toward exploitative control. The 2D t-SNE projection (Fig. 7c) separates the $D_1$ latent space by motor performance, where higher motor performance trials form close clusters near the task manifold, whereas lower-performing trials cluster in diffuse regions associated with longer competition time and larger path deviations (RMSD).

We conducted a sensitivity analysis on $D_1$ over window $W \in \{5, 10, 25, 50, 75, 100, 150, 225, 300, 450\}$, decay $\alpha \in \{0.05, 0.1, 0.2, 0.3, 0.5\}$, and $(\beta_1, \beta_2) \in \{0.1, 0.3, 0.5, 0.7, 0.9\}^2$ to identify stable parameters for the E–E metric calculation. Three convergent patterns founded in the chosen configuration:

    i. S-/R-/T-Corr curves rise sharply and plateaued near $W \approx 120$ aligning with a participant's full trial count (Figure 8);

    ii. Normalized E–E varies $< 6\%$ (CV) across $\alpha \in [0.05, 0.3]$, $\beta_1 \in [0.05, 0.2]$, $\beta_2 \in [0.3, 0.9]$. The setting $\alpha = 0.05$, $\beta_1 = 0.10$, $\beta_2 = 0.90$ balances strong early exploration ($\beta_2 \gg \beta_1$) with a smooth decay to a modest baseline ($\beta_1$);

    iii. The MIN distance (minimum Euclidean distance in embedding to any prior trial within the decayed window) consistently outperformed KNN averaging on ranking quality, indicating sharper novelty discrimination (Table 6a).

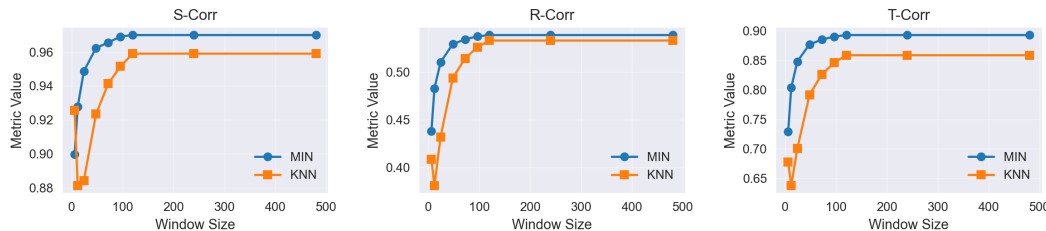

Figure 8: Sensitivity curves comparing MIN vs. KNN. Curves plateau near $W \approx 120$; MIN consistently dominates KNN across decay settings.

Table 6: Novelty metric comparison and clustering diagnostics on SMT-Learner embeddings.

(a) Novelty metric comparison

| Metric variant | ROC AUC | PR AUC | F1 |
|---|---|---|---|
| min_dist (MIN) | **0.5521** | **0.5403** | **0.6994** |
| knn_avg (KNN) | 0.4962 | 0.5026 | 0.6927 |

PR: Precision–Recall; AUC: Area Under the Curve.
min_dist: minimum distance to any prior trial within window $W$
knn_avg: mean distance to the $K$ nearest prior trials

(b) Clustering diagnostics

| Algo | $k/\varepsilon$ | Silhouette | Purity |
|---|---|---|---|
| KMeans | 3 | 0.478900 | 0.962428 |
| KMeans | 5 | 0.405406 | 0.982313 |
| KMeans | 7 | 0.290197 | 0.971298 |
| DBSCAN | 0.5 | 0.361061 | **1.000000** |
| DBSCAN | 1.0 | **0.670152** | 0.999547 |
| DBSCAN | 1.5 | 0.253241 | 0.998204 |

Clustering diagnostics on SMT-Learner embeddings confirmed separability with density-based methods, results in Table 6b. DBSCAN at $\varepsilon = 1.0$ achieves the highest silhouette score with near-perfect purity, reinforcing that the latent geometry supports separable task–performance manifolds. These diagnostics substantiate the parameterization used for downstream E–E estimation.

## A.4 CASE STUDY: OPTIMAL STRATEGY DETECTION

Our framework is capable to detect motor tasks with potentially multiple optimal strategies. We reasoned that the optimal solution to our experimental task was to move to the target in the most efficient path, thereby reducing uncertainty and physiological effort. Optimal solutions could also vary dependent on other environmental conditions (presence of reward, verbal instructions). To provide further clarification, we conducted a case study analysis showing two distinct optimal strategies: (1) Curvature optimization to near-straight paths (mostly used for unimanual), and (2) Stepwise optimal movement with directional changes (mostly used for bimanual). The Table 7 shows the case study results with participants MRTLRN070 and MRTLRN015 (Figure 9 illustrates original trajectories). The E-E framework successfully captured both strategies with a significant E-E ratio reduction (curvature: $0.56 \rightarrow 0.04$, and stepwise: $0.57 \rightarrow 0.07$). Curvature optimization resulted in highly consistent smooth movements (lower final E-E), while stepwise control maintained inherent variability in segmented movements (higher final E-E). This case study demonstrates that SMT-Learner can

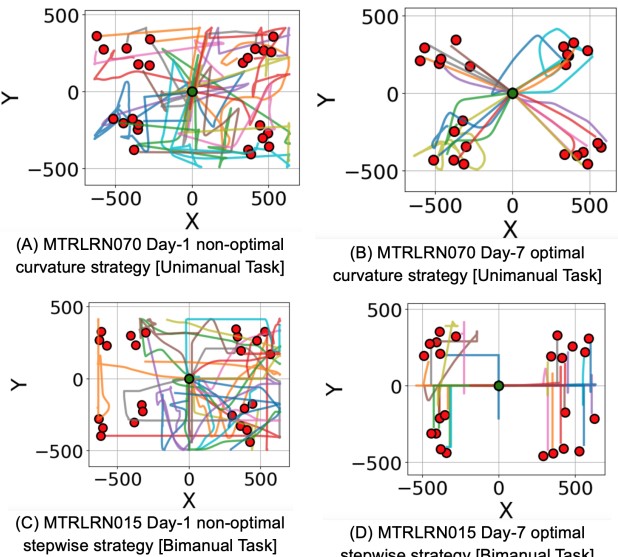

Figure 9: Case study: example of curvature and stepwise optimal movement strategies in motor skill learning

handle multiple optimal strategies in the movement space, enabling quantitative differentiation of strategic signatures.

Table 7: Case study results showing difference between two distinct optimal strategies (participant MTRLRN070: Curvature and MTRLRN015: Stepwise)

| Strategy | E-E Ratio | | Success Rate | | Completion Time (s) | | RMSD | |
|---|---|---|---|---|---|---|---|---|
| | Day-1 | Day-7 | Day-1 | Day-7 | Day-1 | Day-7 | Day-1 | Day-7 |
| Curvature | 0.5579 | **0.0433** | 91.67% | **100%** | 7.34 | **2.36** | 152.74 | **35.84** |
| Stepwise | 0.5718 | 0.0681 | 20.83% | **100%** | 10.19 | 5.10 | 270.91 | 152.85 |

## A.5 Capturing Motor Control Beyond Geometry: SMT-Learner Embeddings

We computed E–E ratios on normalized trajectories and in the learned embedding using $N = 1000$ random samples from $D_1$ Term and Preterm cohorts. As shown in Figure 10, embedding-space E–E yields stable, interpretable effects with tight confidence intervals (CIs), whereas trajectory-space E–E exhibits large-magnitude, high-variance estimates driven by residual geometric/scale variability despite normalization. For early→late learning, the embedding difference is $0.1503$ with a narrow 95% CI $[0.1329, 0.1680]$, while the trajectory estimate is $162.71$ with a very wide CI $[1.79, 441.73]$. For Preterm–Term (bimanual), the embedding difference is $0.0075$ with CI $[-0.0229, 0.0337]$, whereas the trajectory-based mean difference is $-274.75$ with a wide CI $[-1169.24, 112.60]$. These results indicate that SMT-Learner's embeddings capture higher-order control structure beyond geometric variability and provide a scale-stable E–E metric.

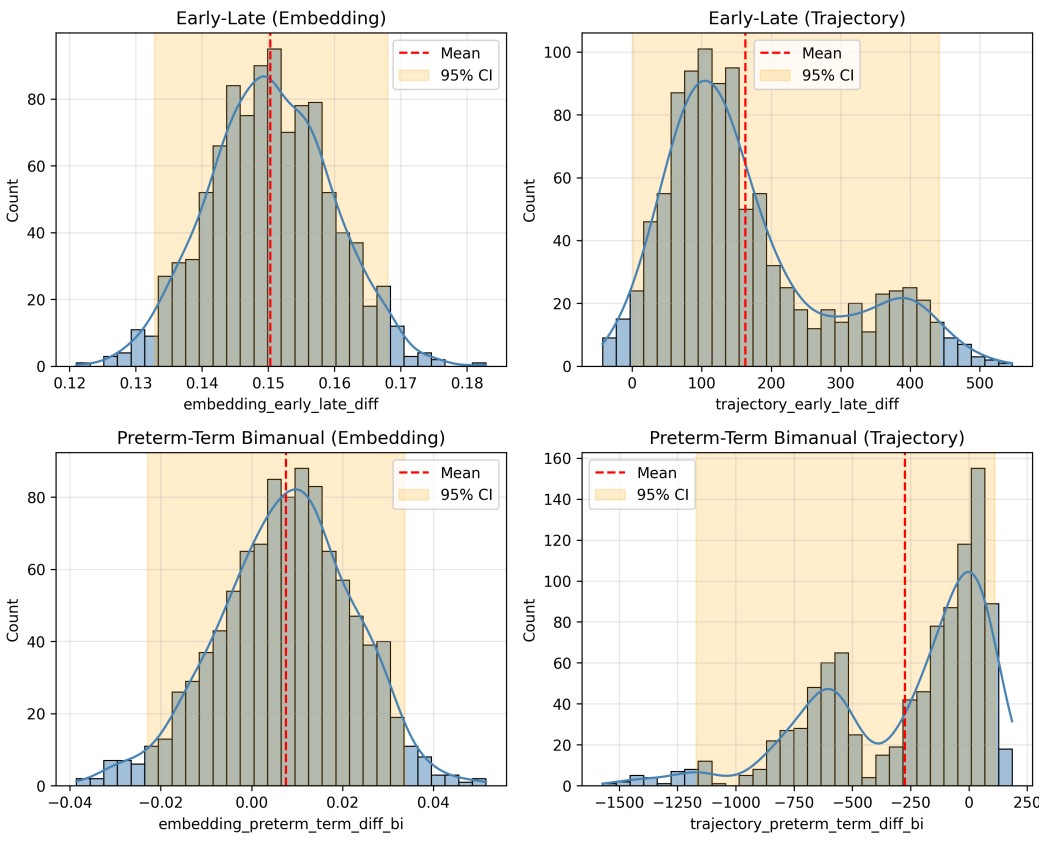

Figure 10: Randomized sampling distributions of E–E differences in embedding space vs. normalized trajectory space ($N = 1000$). Embedding E–E shows tight, stable CIs; trajectory E–E exhibits high variance due to residual geometric/scale effects.

