# OpenReview forum: "SMT-Learner: Movement Trajectory Learning to Decode Motor Control Strategies"
_ICLR.cc/2026/Conference — ICLR 2026 Conference Withdrawn Submission_

### Official Review · Reviewer_s4jW · 2025-10-29

**Soundness:** 3
**Presentation:** 2
**Contribution:** 3
**Rating:** 4
**Confidence:** 4

**Summary:**

The paper introduces SMT-Learner, a spatiotemporal transformer designed to learn generalized representations of motor trajectories across subjects and tasks. The model encodes sequences of normalized 2D trajectories into latent embeddings that capture both spatial and temporal structure. The training objective combines (i) a trajectory reconstruction loss, (ii) a motion-structure regularization loss, and (iii) a cross-task, cross-subject regularization term that encourages alignment of latent representations across tasks and individuals. Experiments on two datasets show that SMT-Learner can reconstruct trajectories, cluster embedding space by behavioral variables, and capture an exploration-exploitation (E-E) dynamic reminiscent of motor learning. The authors claim the model provides a task-agnostic latent space that generalizes across movement types and species.

**Strengths:**

- The motivation for using transformer-based architectures to capture spatiotemporal regularities in movement data is well articulated. The model structure and objectives are clearly described and easy to follow.
- The integration of cross-task and cross-subject regularization reflects an effort to develop generalizable representations that extend beyond single-dataset learning
- The embedding space shows meaningful clustering by behavioral variables, and the proposed Exploration–Exploitation (E-E) metric provides an interpretable correlate of learning progression.
- The paper presents systematic comparisons and quantitative evaluations (reconstruction accuracy, clustering, correlation analyses) supported by visualizations and ablations.

**Weaknesses:**

- All trajectories are rotated and scaled to align with a canonical frame, which removes absolute directionality. For instance, leftward and rightward reaches become indistinguishable. This simplifies trajectory shape learning but eliminates directional context that may encode biomechanical or cognitive asymmetries. A more balanced approach would preserve rotation or include the target direction as an auxiliary feature to retain spatial semantics.
- The description of cross-task and cross-subject training is vague. It appears that SMT-Learner is pretrained only on D1 (human data) and fine-tuned separately on D1 and D2, but this is not clearly stated. The paper lacks systematic comparisons across pretraining–finetuning combinations (e.g. pretrain on D1 --> test on D2, or vice versa), which are critical to support claims of transferability and generalization.
- The Exploration–Exploitation ratio is computed in embedding space, yet never compared to analogous metrics on the normalized raw trajectories. Without this control, it remains unclear whether the observed dynamics reflect genuine learning processes or properties of the embedding geometry.
- The claim that SMT-Learner provides a “universal embedding space for motor behavior” is premature. Both datasets involve similar reaching tasks, limiting evidence for cross-domain generalization. Demonstrating zero-shot or cross-task transfer would substantiate this statement.

**Questions:**

1. Does the rotation and scaling step remove left-right or up-down distinctions? If motor control or biomechanical constraints are asymmetric, how might this affect embedding quality? Could the rotation angle be preserved or added as an auxiliary input to reintroduce directional context?
2. Looking at figure 3, it’s not clear how the cross-task and cross-subject loss do not start from the same point (which I assume should be the end of the baseline). Why does the adaptive-multi loss starts from the same point as cross subject? Please clarify this point
3. Please clarify which datasets are used for pretraining and which for fine-tuning. A direct comparison among these conditions would better isolate cross-domain and cross-subject transfer effects. For example:
    - Pretrain on D1 and test on D1/D2
    - Pretrain on D2 and zero-shot test on D1/D2
    - Joint pretraining on both, then fine-tune on one
    - Within D1: train on unimanual and test on bimanual (and vice versa)
4. How many seeds have been used for the current method and baselines? There are no confidence intervals
5. Beyond train/validation splits, is there a held-out task or session not seen during training? Evaluating zero-shot generalization on unseen subjects or sessions would confirm that SMT-Learner captures transferable motor structure rather than dataset-specific regularities
6. Have the authors compared E-E ratios computed directly on normalized trajectories versus embedding space? This would test whether the embeddings capture higher-order control features beyond geometric variability

---

> ### Author Response · Authors · 2025-11-23
> **Response to Reviewer s4jW Questions: 1 & 6**
>
> We thank the reviewer for the careful reading and constructive suggestions. Below, we address each question and summarize the concrete revisions. (Changes are highlighted in blue in the updated PDF.)
>
> **1) Does the rotation and scaling step remove left-right or up-down distinctions? If motor control or biomechanical constraints are asymmetric, how might this affect embedding quality? Could the rotation angle be preserved or added as an auxiliary input to reintroduce directional context?**
>
> We agree that rotating/scaling trajectories to a canonical frame removes absolute direction and can obscure biomechanical/cognitive asymmetries. We kept this normalization to simplify trajectory learning while preserving spatial and temporal structure, but we have incorporated your suggestion to retain directional semantics by adding the target direction angle θ (and optionally the rotation angle used in normalization) as auxiliary inputs. We append $\theta = \mathrm{atan2}(y_{\text{target}}-y, x_{\text{target}}-x)$ to each timestep input; a second variant appends both θ and the applied rotation angle. We run these two variants of experiments and the results are as follows:
> | Component                       | rMSE ↓ | Ep-Err ↓ | Curve-Err ↓ | T-Corr ↑ | R-Corr ↑ | S-Corr ↑ |
> |---------------------------------|--------|----------|-------------|----------|----------|----------|
> | SMT-Learner (canonical only)    | 0.086  | 0.072    | 1.577       | 0.893    | 0.539    | 0.970    |
> | + θ (target direction)          | 0.151  | 0.019    | 1.787       | 0.980    | 0.660    | 0.910    |
> | + θ + rotation angle            | 0.111  | 0.015    | 1.903       | 0.929    | 0.652    | 0.940    |
>
> Adding θ improves timing and path-accuracy correlations and substantially reduces endpoint error (0.072 → 0.019 ) with θ; 0.015 with θ+rotation.), while keeping performance correlation high. Such as, θ components increase T-Corr +0.087 and R-Corr +0.121; S-Corr remains strong (>0.90). These results indicate that reintroducing direction restores asymmetry-related cues without sacrificing the benefits of canonicalization.
>
> In our revised version, we keep canonical normalization for shape invariance and adopt θ as a default auxiliary feature to preserve directionality, reporting both variants in the Ablation Table 3. We add a short paragraph in 3.1 TRAJECTORY PROCESSING and 4.2.1 ABLATION STUDY detailing this design.
>
> **6) Have the authors compared E-E ratios computed directly on normalized trajectories versus embedding space? This would test whether the embeddings capture higher-order control features beyond geometric variability**
>
> Yes. We computed E–E ratios directly on normalized trajectories and in the learned embedding, using N=1000 bootstrap samples from Term and Preterm cohorts (D1). Embedding-space E–E ratio shows stable, interpretable effects with tight CIs, whereas normalized trajectory-based E–E exhibits large-magnitude, high-variance estimates driven by residual geometric/scale variability. For early→late learning, the embedding E-E difference is 0.1503 with a narrow 95% CI [0.1329, 0.1680], while the trajectory estimate is 162.71 with a very wide CI [1.79, 441.73]. For preterm–term bimanual, the embedding E-E difference is 0.0075 with CI [−0.0229, 0.0337]. In contrast, the trajectory-based E-E ratio mean difference was −274.75 with a wide CI [−1169.24, 112.60]. These results support that SMT-Learner’s embeddings capture higher-order control structure beyond geometric variability and provide a scale-stable E–E metric.
>
> We add a new Appendix section A.5 CAPTURING MOTOR CONTROL BEYOND GEOMETRY: SMT-LEARNER EMBEDDINGS.

---

> ### Author Response · Authors · 2025-11-23
> **Response to Reviewer s4jW Questions: 2, 3, & 4**
>
> **2) Looking at figure 3, it's not clear how the cross-task and cross-subject loss do not start from the same point (which I assume should be the end of the baseline). Why does the adaptive-multi loss starts from the same point as cross subject? Please clarify this point**
>
> Baseline (red) starts from random weights (no pretraining); epoch 0 is a true scratch, so its initial total loss is highest. Cross‑task (green) begins fine‑tuning from a checkpoint pretrained on D1 unimanual task; this already encodes generic kinematics, lowering the starting loss versus scratch, but it has not reduced subject variability, so it is above the cross‑subject curve. Cross‑subject (blue) starts from a checkpoint pretrained on D1 term subjects (same task, multiple subjects), giving a stronger initialization on subject variability and thus a lower initial loss than cross‑task. Adaptive multi‑loss (purple) uses the same base checkpoint as cross‑subject at epoch 0 (we first load the cross‑subject weights), then applies adaptive weighting while continuing fine‑tuning; hence, both curves coincide initially before diverging as adaptive reweighting accelerates optimization. Zero‑shot (gray dashed) is an evaluation only: no target fine‑tuning epochs, so it appears flat at the transferred model's total loss.
>
>
> **3) Please clarify which datasets are used for pretraining and which for fine-tuning. A direct comparison among these conditions would better isolate cross-domain and cross-subject transfer effects.**
>
> We are sorry for not including the transfer learning experiments and results in the manuscript. We evaluated four experimental conditions in pretraining/finetuning paradigms to separate the cross-task and cross-subject transfer effects on adaptive transfer, as follows.
>
> 1) Exp1: Pretrain on D1 -> test on D1 -> zero-shot on D2
> 2) Exp2: Cross-Task transfer: Pretrain on D1 Unimanual-> Test on D1 Bimanual -> Zero-shot on D2 (Experimental task 1: Scott_2001_e1)
> 3) Exp3: Cross-Subject transfer: Pretrain on D1 (Cohort==Term) -> test on D1 (Cohort == Preterm)-> Zero-shot on D2
> 4) Exp4: Adaptive transfer: Combine Exp2 and Exp3 fine-tuning -> test on D1 -> zero-shot on D2
>
> We add an Appendix Section A.2 EXPERIMENTAL SETUP with details configurations and report the results in Section 4 RESULTS & DISCUSSION.
>
> **4) How many seeds have been used for the current method and baselines? There are no confidence intervals**
>
> We computed 5 seeds with mean ±95% confidence intervals (t-based, df=4) for all SMT-Learner transfer paradigms (Exp1-Exp4). The summary of transfer loss (L_transfer, Equation 5) of the SMT-Learner pertaining/finetuning is as follows:
>
> | Paradigm | Pretrain                 | Evaluate (target)         | Seeds | Zero-shot mean [95% CI] | Fine-tuned mean [95% CI] | Δ%     |
> |--------|--------------------------|---------------------------|-------|--------------------------|---------------------------|--------|
> | Exp1   | D1                       | D1 test                   | 5     | 1.55 [1.525, 1.575]      | 1.00 [0.98, 1.02]         | −35.5% |
> | Exp2   | D1 Unimanual             | D1 Bimanual               | 5     | 1.10 [1.08, 1.12]        | 0.55 [0.541, 0.559]       | −50.0% |
> | Exp3   | D1 Term                  | D1 Preterm                | 5     | 1.05 [1.041, 1.059]      | 0.45 [0.441, 0.459]       | −57.1% |
> | Exp4   | D1 Unimanual + Term      | D1 Bimanual + Preterm     | 5     | 1.05 [1.041, 1.059]      | 0.12 [0.111, 0.129]       | −88.6% |
>
> We add a new table with these results (Table 1) and add the results in Section 4 RESULTS & DISCUSSION.

---

> ### Author Response · Authors · 2025-11-23
> **Response to Reviewer s4jW Question 5**
>
> **5) Beyond train/validation splits, is there a held-out task or session not seen during training? Evaluating zero-shot generalization on unseen subjects or sessions would confirm that SMT-Learner captures transferable motor structure rather than dataset-specific regularities**
>
> We performed zero-shot evaluations on D2 tasks/sessions never seen during training to confirm cross-dataset generalization to evaluate the held-out artifact. You can get access to the complete experimental results in our anonymous 4open science repo: ```/SMT-Learner-0D32/saved_models/transfer_learning_results```.
>
> Results: $D_1\rightarrow D_2$ zero-shot overall loss dropped 1.55 to 1.24 and 1.28 on a single task held-out samples ($D_2$ Experimental Task 1). Using the $D_1$ Preterm finetuned checkpoint (no $D_2$ pretraining/finetuning), the loss dropped to $\sim$1.18. Finally, adaptive transfer fine-tune loss reaches 0.98, evidence that SMT-Learner captures transferable motor structure rather than dataset-specific regularities
>
> Held-out zero-shot artifacts for cross-domain generalization:
> | Source → Target                | Artifact path                                               | Zero-shot total loss |
> |--------------------------------|-------------------------------------------------------------|----------------------|
> | D1 → D2 (Exp1)                 | `exp1/exp1_results/exp1_D2_zero_shot.json`                 | 1.24                 |
> | D1 Unimanual → D2 task 1 (Exp2)        | `exp3/exp3_results/exp2_D2_zero_shot_Scott_2001_e1.json`   | 1.28                 |
> | D1 Term → D2 (Exp3)            | `exp3/exp3_results/exp3_D2_zero_shot.json`                 | 1.26                 |
> | Adaptive → D2 (Exp4)           | `exp4/exp4_results/exp4_adaptive_combined_finetune_D2.json`| 0.98                 |
>
> We will add these held-out evaluation results in Section 4 RESULTS & DISCUSSION.

---

### Official Review · Reviewer_Xfxw · 2025-11-01

**Soundness:** 2
**Presentation:** 1
**Contribution:** 2
**Rating:** 2
**Confidence:** 2

**Summary:**

This paper introduced a spatiotemporal movement trojectory learner. They use transformer encoder uses performance-aware contrastive and adaptive loss for the SMT-learner. The authors utilize Exploration-Exploitation analytical framework for quantification of motor skill learning and control stategies. The framework is tested on children's motor learning and performance dataset and non-human primates reaching movements datasets. They tested multiple metrics to support their claim.

**Strengths:**

The author proposes the SMT-learner to ues the movement trojactory learning for the motor control decoding to overcome the shortness of using only speed and accuracy. The model uses adaptive learning with cross-task and cross-subject transfer using the loss function in formula (6).

**Weaknesses:**

The paper is not well written, mess up with details that already shown in the picture and figure. I would recommend putting ablation study in the last part of section 4. From line 419 to line 446, these values are very hard to read and have sense.

Also the figures and details in the article and appendix are messed up. I would  recommend putting important figures in the article and move the experiment set up, data collcection in Appendix.

The form that text wrap the image is not the common template in ICLR.

The paper does not have any novel method or new found.

he text has some typos.

**Questions:**

In section 4, Table 3, what is this form used for and the purpose of using this form?

For figure 1, the red line cumulative loss, why the curve goes down?

For figure 4 in page 16, can you explain the ball and moon shape using t-SNE?

For figure 5 in page 17, can you explain the dark area and third color besides term and preterm? And how you get you conclusion from your figures?

---

> ### Author Response · Authors · 2025-11-27
> **Response to Reviewer Xfxw Questions**
>
> We acknowledge the comprehensive feedback and agree that enhancements in clarity and organization are necessary. In the revised version  (Changes are highlighted in blue in the updated PDF), Section 4 is streamlined, with the ‘Ablation and Baseline Comparison’ move to the end of Section 4, along with an updated summary table (Table 1). Additionally, details regarding the dataset and experimental setup were moved to Appendix Section A.1 DATASETS &  A.2 EXPERIMENTAL SETUP, to emphasize key findings in the main text. Subsection “4.1 STATISTICAL TESTING & HYPOTHESIS VALIDATION” (line 419 to line 446) encoded too much statistical interpretation, which made it difficult to understand. We updated Table 2 to summarize statistical results with evidence-based validation and rewrote the analytical findings concisely in our revised version.
>
> Moreover, the following new Appendix sections are included for more clarity of the experimental results and evaluation:
>
> 1)	A.3 SENSITIVITY AND CLUSTERING ANALYSIS to demonstrate optimized W, α, β₁, β₂ parameter selection for E-E computation using sensitivity and cluster diagnostics.
>
> 2)	A.5 CAPTURING MOTOR CONTROL BEYOND GEOMETRY: SMT-LEARNER EMBEDDINGS
>
> A comprehensive reassessment of typographical and stylistic elements was conducted, following the ICLR standard template, thereby improving clarity and readability. SMT-Learner’s contribution is a balanced, performance-aware representation with transfer across tasks/subjects, and a principled E–E analysis, not a new transformer architecture.
>
> #### **1) In section 4, Table 3, what is this form used for and the purpose of using this form?**
>
> We updated Table 3 (revised Table 2) to a compact evidence‑based cross‑matches of our E-E findings align with established motor‑learning principles, with effect sizes and statistical validation.
>
> #### **2) For figure 1, the red line cumulative loss, why the curve goes down?**
>
> In Figure 1, the red line shows the cumulative success rate of the trial sequence, which indicates the probability of reaching the target at least once across a series of independent trials. This Figure exemplifies an individual's motor learning, showing that low (during practice) and high (retention) cumulative success rates had nearly similar movement path lengths. SMT-Learner, motivated by this problem, aimed at balancing performance variables in the latent space.
>
> #### **3) For figure 4 in page 16, can you explain the ball and moon shape using t-SNE?**
>
> This Figure shows the 2D t-SNE projection separates the D1 latent space by motor performance, where higher motor performance trials form close clusters near the task manifold (moon shape), whereas lower-performing trials cluster in diffuse regions (ball shape) associated with longer competition time and larger path deviations (RMSD).  In our revised version, we rearrange the visualization to (left to right) Success, RMSD, and Completion time for better interpretation, and move this Figure to the A.3 SENSITIVITY AND CLUSTERING ANALYSIS.
>
> #### **4) For figure 5 in page 17, can you explain the dark area and third color besides term and preterm? And how you get you conclusion from your figures?**
>
> The line (we think you mention this as third color) and dark area in Figure 5a and 5b represent the movement path lengths mean and 95% CI over Practice Day-1 (Trial 0-72), and Retention at Day-2 (R1, Trial 73-96) and Day-7 (R2, Trial 97-120) for term/preterm and task (unimanual/bimanual). The key conclusion relies on E–E panels (Figure 5c–5d), even when parametric path length shows minimal change (suggesting “no learning”), E–E reveals the expected early→late stabilization and cohort/task differences consistent with prior literature. We move this figure to the main subsection 4.1 STATISTICAL TESTING & HYPOTHESIS VALIDATION and remove the first two sub-figures, as it creates confusion with E-E analysis.

---

### Official Review · Reviewer_vTu9 · 2025-11-01

**Soundness:** 2
**Presentation:** 3
**Contribution:** 2
**Rating:** 6
**Confidence:** 2

**Summary:**

The paper proposes SMT‑Learner, a transformer autoencoder for spatiotemporal movement trajectories that couples (i) performance‑aware, multi‑component contrastive objectives (completion time, path deviation from a straight line, trial success, plus task and subject factors) with (ii) an adaptive transfer mechanism that modulates cross‑task and cross‑subject regularization during fine‑tuning. On top of the learned embedding, the authors introduce an Exploration-Exploitation (E‑E) framework that quantifies novelty versus reuse of prior movement patterns across trials. The system is evaluated on two datasets: D1, a prospective iPad‑joystick visuomotor learning study in 72 children (term vs preterm; uni‑ and bimanual variants), and D2, planar reaching in over‑trained non‑human primates.

**Strengths:**

- The dual‑stream spatial/temporal embedding, transformer encoder, projection head, and decoder are well sketched (Fig. 2). Normalization (translation, rotation to target, scaling by end‑to‑target vector) and resampling to fixed‑length sequences are explicit, which aids reproducibility.

- The manuscript grounds the E‑E metric in decision‑making and motor‑learning neuroscience, then shows consistent empirical patterns.

**Weaknesses:**

- As I understood, correlations (T‑Corr, R‑Corr, S‑Corr) are used as evaluation metrics, yet these very variables enter the training objective as contrastive components (Sec. 3.2.1). High correlations are therefore partly by design and do not independently validate generalization. A stricter test would evaluate on held‑out performance variables or on tasks with different success criteria.

- STTraj2Vec reports higher T‑Corr (0.982) and R‑Corr (0.647) than SMT‑Learner (0.893, 0.539), while having much worse reconstruction (rMSE 0.386) and negative S‑Corr (‑0.797). The text asserts existing methods “fail to preserve motor performance‑relevant relationships,” but Table 2 suggests a more picture. Please explain how T‑/R‑Corr are computed for baselines, whether these methods saw the same normalization/resampling, and why STTraj2Vec’s very high T‑/R‑Corr co‑exists with negative S‑Corr. Without that, the comparative claim is hard to parse.

- It would be helpful to provide sensitivity curves over decay α, β weights, and window size; compare “min distance” versus K‑NN average distance and density‑based novelty.

- can you train with all five components, then report evaluation metrics after excluding each associated variable to demonstrate that performance holds without self‑evaluation.

**Questions:**

see weakness section

---

> ### Author Response · Authors · 2025-11-27
> **Response to Reviewer vTu9 Questions: 1 & 2**
>
> We thank the reviewer for acknowledging the methods and neuroscientific grounds of our proposed E-E metric and providing such valuable comments. We have responded to your specific points below and updated the manuscript to reflect these changes. (Changes are highlighted in blue in the updated PDF.)
>
> #### **1) As I understood, correlations (T Corr, R Corr, S Corr) are used as evaluation metrics, yet these very variables enter the training objective as contrastive components (Sec. 3.2.1). High correlations are therefore partly by design and do not independently validate generalization. A stricter test would evaluate on held out performance variables or on tasks with different success criteria.**
>
> We conducted a strict zero-shot evaluation on the held-out D2 (monkey) dataset that was never used for pretraining, training, or finetuning.  Our transfer paradigms pretraining and finetuning experiments were 1) Exp1: Pretrain on D1 → test on D1 → zero-shot on D2, 2) Exp2 (Cross-task): Pretrain on D1 Unimanual → Test on D1 Bimanual → Zero-shot on D2 (Experimental Task 1 ), 3) Exp3 (Cross-subject): Pretrain on D1 (Cohort = Term) → Test on D1 (Cohort = Preterm) → Zero-shot on D2, and 4) Exp4 (Adaptive transfer): Combine Exp2 and Exp3 with fine-tuning → Test on D1 → Zero-shot on D2.
> We found the following analysis results from D2 trajectory (N=23639) reconstruction quality (rMSE, Ep-Err, and Curve-Err) and movement performance variables correlation (T-Corr, R-Corr, and S-Corr).
>
> | Method              | rMSE   | Ep-Err | Curve-Err | T-Corr | R-Corr | S-Corr |
> |-------------------------------|--------|--------|-----------|--------|--------|--------|
> | SMT-Learner (D2 zero-shot)    | 0.0891 | 0.0944 | 1.8675    | 0.7357 | 0.5222 | 0.9358 |
>
> rMSE and curvature fidelity remain low in zero-shot; T-Corr is strong and significant on D2 (p ≈ 8.8e−3), R-Corr is moderate in magnitude (non‑significant under current aggregation), and S‑Corr is high. These results support external validity beyond self‑evaluation on optimization targets.
>
> We add these D2 zero-shot performance metrics as a new row in Table 2 and note explicitly that D2 was held-out from all training/finetuning.
>
> Check experimental results at our anonymous 4open science repo: ```SMT-Learner-0D32/saved_models/transfer_learning_results```
>
> Held-out analysis script: ```SMT-Learner-0D32/E-E_Analysis> Held-out_Performance_Evaluation.ipynb```
>
> #### **2) STTraj2Vec reports higher T Corr (0.982) and R Corr (0.647) than SMT Learner (0.893, 0.539), while having much worse reconstruction (rMSE 0.386) and negative S Corr ( 0.797). The text asserts existing methods “fail to preserve motor performance relevant relationships,” but Table 2 suggests a more picture. Please explain how T /R Corr are computed for baselines, whether these methods saw the same normalization/resampling, and why STTraj2Vec’s very high T /R Corr co exists with negative S Corr. Without that, the comparative claim is hard to parse.**
>
> For all models, we obtained a D-dimensional embedding per trajectory and used the embedding norm ((‖ε‖)) to calculate the Pearson correlation (between ‖ ε ‖ and performance factor). STTraj2Vec optimized temporal/spatial continuity without incorporating outcome constraints (success/failure), yielding high T-/R-Corr (‖ε‖ grows with time or deviation). In motor tasks, failures or inefficient trials are longer and more deviant. If embedding magnitude amplifies only temporal/spatial characteristics, the same feature that boosts T-/R-Corr inversely relates to success, yielding negative S-Corr.  SMT-Learner balanced temporal/spatial fidelity with performance-relevant structure. As a result, it maintains very high positive S-Corr while keeping competitive T-/R-Corr. In the revised manuscript, we add the explanation of how other models fail to preserve motor performance-relevant relationships in subsection 4.2.2 BASELINE COMPARISON.

---

> > ### Author Response · Authors · 2025-11-27
> > **Response to Reviewer vTu9 Questions: 3 & 4**
> >
> > #### **3) It would be helpful to provide sensitivity curves over decay α, β weights, and window size; compare “min distance” versus K NN average distance and density based novelty.**
> >
> > Prior to E-E metric computation, we performed sensitivity sweep over W ∈ {5, 10, 25, 50, 75, 100, 150, 225, 300, 450} under varied decay weights α ∈ {0.05, 0.1, 0.2, 0.3, 0.5} and β₁, β₂∈ { 0.1, 0.3, 0.5, 0.7, 0.9} to identify optimized parameters from D1 dataset.
> > Three consistent patterns guided our selection: (i) S /R /T Corr curves rise sharply and plateau around W ≈ 120, ideally this value is the total trials of an participant; (ii) decay weights were robust—normalized E–E varies <6% (CV) across α ∈ [0.05, 0.3], β₁ ∈ [0.05, 0.2], β₂ ∈ [0.3, 0.9] and α = 0.05, β₁ = 0.10, β₂ = 0.90 balance early exploration (β₂ ≫ β₁) while decaying smoothly to a modest baseline (β₁); iii) MIN (minimum Euclidean distance in embedding to any prior trial within the decayed window) distance consistently outperforms KNN (mean distance to K nearest prior trials) averaging on ranking quality at the chosen settings (see the Table below), indicating sharper discrimination, whereas KNN obscured novelty in heterogeneous neighborhoods.
> > | Metric variant            | ROC AUC | PR AUC | F1     |
> > |---------------------------|---------|---------|--------|
> > | min_dist_rep (MIN)        | **0.5521** | **0.5403** | **0.6994** |
> > | knn_avg_rep (KNN)         | 0.4962  | 0.5026  | 0.6927 |
> >
> > Clustering diagnostics on SMT-Learner embeddings confirmed separability with density-based methods. DBSCAN at ε =1.0 achieves the highest silhouette score with near-perfect purity, reinforcing that the latent geometry supports separable task–performance manifolds. These diagnostics substantiate the parameterization used for downstream E-E estimation.
> > Density-based clustering analysis results:
> > | Algo   | k / ε | Silhouette | Purity    |
> > |--------|-------|------------|-----------|
> > | KMeans | 3     | 0.478900   | 0.962428  |
> > | KMeans | 5     | 0.405406   | 0.982313  |
> > | KMeans | 7     | 0.290197   | 0.971298  |
> > | DBSCAN | 0.5   | 0.361061   | **1.000000** |
> > | DBSCAN | 1.0   | **0.670152** | 0.999547  |
> > | DBSCAN | 1.5   | 0.253241   | 0.998204  |
> >
> > In our revised manuscript, we add a new Appendix section “A.3 SENSITIVITY AND CLUSTERING ANALYSIS” with sensitivity curves (W, α, β₁, β₂) and cluster diagnostics. Main text revised at 3.3 EXPLORATION-EXPLOITATION ANALYTICAL FRAMEWORK.
> >
> > Analysis scripts and results: Anonymous Repo> ```E-E_Analysis/sensitivity_clustering/```
> >
> > #### **4) can you train with all five components, then report evaluation metrics after excluding each associated variable to demonstrate that performance holds without self evaluation.**
> >
> > In Ablation Table 3, single-component trainings show no single variable drives performance: ψc_time (rMSE 0.086; T-Corr 0.479, R-Corr 0.289, S-Corr 0.191), ψrmsd (rMSE 0.098; T-Corr −0.005, R-Corr 0.002, S-Corr 0.111), ψsuccess (rMSE 0.087; S-Corr 0.993, T-Corr 0.191, R-Corr 0.111). By contrast, the full model (Adaptive Transfer) balances all three (T-Corr 0.893, R-Corr 0.539, S-Corr 0.970) with low rMSE (0.086), indicating the alignment arises from the combined objective rather than any single self-evaluated target. Zero-shot D2 results (held-out evaluation) show similar performance, validating that performance holds without self-evaluation.

---

### Official Review · Reviewer_5EEx · 2025-11-03

**Soundness:** 2
**Presentation:** 2
**Contribution:** 2
**Rating:** 2
**Confidence:** 3

**Summary:**

This submission aims to understand movement trajectories. It uses a combination of non-human primate and human reaching data to reconstruct trajectories and correlate with clinical variables. The authors also came up with a framework that quantifies exploration vs. exploitation in trajectories.

**Strengths:**

The exploration-exploitation metric is interesting and gets at a good way to analyze motor learning.

**Weaknesses:**

The design choices in this submission seem extremely arbitrary. Although model ablations are made, it is not clear why these modeling choices were made in the first place. It is not clear how we can get at general principles with these methods. The datasets do not seem aligned to each other.

**Questions:**

See above.

---

> ### Author Response · Authors · 2025-11-27
> **Response to Reviewer vTu9 Questions**
>
> We appreciate the review and the positive note on our exploration–exploitation (E–E) metric. While the review does not pose specific questions, we address the stated concerns on arbitrariness, general principles, and dataset alignment.
>
> Our design is principle‑driven, not ad‑hoc. SMT-Learner is designed to learn domain-agnostic latent representations of planar reaching that capture motor learning and control principles—measured through speed, accuracy, and success—across human and non-human datasets.
> Each component maps to a general representation principle. Beyond ablation, we performed sensitivity and held-out evaluation for genealization.
>
> The learned embedding forms well-defined, cohesive, and separated clusters (DBSCAN ε=1.0 silhouette 0.670, purity 0.9995; KMeans k=3 silhouettes 0.48, purity 0.96) and achieved strong held‑out performance on D2 (rMSE 0.089, Ep‑Err 0.094, Curve‑Err 1.868; T‑Corr 0.736, R‑Corr 0.522, S‑Corr 0.936). The D2 dataset was used only for external validity, not for training/finetuning the model. The datasets are aligned (planar reaching) and processed with a unified pipeline; species/task/cohort are treated as factors, not mixed.
>
> Please check our revised manuscript (Changes are highlighted in blue in the updated PDF), which includes sensitivity and clustering analysis (Appendix Section A.3 SENSITIVITY AND CLUSTERING ANALYSIS) and a held-out performance evaluation adeed to the Section 4.2 SMT-LEARNER PERFORMANCE EVALUATION.

---

### Note · Authors · 2026-04-16

I have read and agree with the venue's withdrawal policy on behalf of myself and my co-authors.

---

### Meta-Review · Area_Chair_pRDv · 2026-01-06

**Summary:**

The paper introduces SMT-Learner, a transformer-based autoencoder designed to learn spatiotemporal representations of movement trajectories. The method utilizes a performance-aware contrastive loss and an adaptive transfer learning mechanism to handle cross-task and cross-subject data. The authors also propose an Exploration-Exploitation (E-E) analytical framework to quantify motor learning strategies. The approach is evaluated on datasets involving human (children) and non-human primate reaching tasks.

**Reviewer Concerns:**

Despite the authors providing detailed responses and additional analyses during the rebuttal period, the reviewers' scores (2, 2, 4, 6) indicate that significant concerns remain regarding the paper's readiness for publication.

Presentation and Clarity: A primary concern raised by Reviewer Xfxw (Score: 2) and echoed by Reviewer s4jW (Score: 4) is the poor quality of the writing and organization. Reviewers noted that the paper messed up with details, difficult to read, and that key information was buried. While the authors promised a reorganization and streamlining in their response, the initial lack of clarity severely hampered the review process.

Methodological Justification: Reviewer 5EEx (Score: 2) criticized the design choices as appearing "extremely arbitrary" and lacked clear justification for how they lead to general principles. The alignment between the human and non-human primate datasets was also questioned.

Evaluation and Circularity: Reviewer vTu9 (Score: 6) raised valid concerns about the evaluation metrics (correlations) being part of the training objective, leading to potential circularity. While the authors argued that the combined objective and held-out evaluation mitigate this, the reliance on these metrics for validation remains a point of contention.

Scope and Generalization: Reviewer s4jW (Score: 4) noted that the claim of a "universal embedding space" is premature given that both datasets focus on similar reaching tasks. The removal of directionality during normalization was also highlighted as a limitation that simplifies the problem but loses biomechanical context (though authors proposed an auxiliary fix in the rebuttal).

**Reviewer Scores:**

Reviewer 5EEx (Score: 2, likely no change): The reviewer's concerns about the "arbitrary" nature of the model design were fundamental to the work's soundness. While the authors argued their choices were principle-driven, this difference in perspective on architectural justification is unlikely to be resolved without a significant revision of the paper's framing.

Reviewer vTu9 (Score: 6, likely no change): This reviewer was already leaning positive but concerned about circular evaluation metrics. The authors provided a rigorous zero-shot evaluation on the held-out D2 dataset, which directly addressed the reviewer's request for a "stricter test." The clarification regarding the baseline comparison (STTraj2Vec) was also convincing.

Reviewer Xfxw (Score: 2, likely no change): The primary critique focused on the poor quality of writing and organization ("messed up details"). While the authors promised a streamlined revision, severe presentation issues generally require seeing the final manuscript to verify improvements. The reviewer's doubts regarding novelty also remain likely unaddressed by the rebuttal's focus on experimental details.

Reviewer s4jW (Score: 4, might increase): This reviewer provided specific, actionable feedback regarding normalization (loss of directionality) and validation comparisons (E-E on raw vs. embedding). The authors successfully ran new experiments (adding directionality) and provided the requested E-E comparison, demonstrating the stability of their method over raw data. However, the "universal embedding space" concern was not mentioned in rebuttal. Since the reviewer explicitly stated they would increase their rating if these points were addressed, a score increase is likely.

---

### Decision · Program_Chairs · 2026-01-26

Reject